# Elastic properties and tensile strength of 2D Ti$_3$C$_2$T$_x$ MXene monolayers

Chao Rong [1,2,3], Ting Su[1,2,3], Zhenkai Li[1,2,3], Tianshu Chu[1,2,3], Mingliang Zhu [1,2,3], Yabin Yan [1,2,3] ✉, Bowei Zhang [1,2,3] ✉ & Fu-Zhen Xuan [1,2,3] ✉

Two-dimensional (2D) transition metal nitrides and carbides (MXenes), represented by Ti$_3$C$_2$T$_x$, have broad applications in flexible electronics, electromechanical devices, and structural membranes due to their unique physical and chemical properties. Despite the Young's modulus of 2D Ti$_3$C$_2$T$_x$ has been theoretically predicted to be 0.502 TPa, which has not been experimentally confirmed so far due to the measurement is extremely restricted. Here, by optimizing the sample preparation, cutting, and transfer protocols, we perform the direct in-situ tensile tests on monolayer Ti$_3$C$_2$T$_x$ nanosheets using nanomechanical push-to-pull equipment under a scanning electron microscope. The effective Young's modulus is 0.484 ± 0.013 TPa, which is much closer to the theoretical value of 0.502 TPa than the previously reported 0.33 TPa by the disputed nanoindentation method, and the measured elastic stiffness is ~948 N/m. Moreover, during the process of tensile loading, the monolayer Ti$_3$C$_2$T$_x$ shows an average elastic strain of ~3.2% and a tensile strength as large as ~15.4 GPa. This work corrects the previous reports by nanoindentation method and demonstrates that the Ti$_3$C$_2$T$_x$ indeed keeps immense potential for broad range of applications.

Two-dimensional (2D) transition metal carbides and nitrides, known as MXenes, are an emerging class of 2D layered materials that have attracted widespread attention due to their excellent metal conductivity[1], hydrophilic properties[2], dispersion stability[3], and flexibility[4]. Since the first MXene (Ti$_3$C$_2$T$_x$) was discovered in 2011 by Yury, et al. [5], the cross combination of physical and chemical properties of it has facilitated extensive investigations on various applications including flexible electronics[6,7], supercapacitors[8], catalysis[9], sensors[10], aerospace[11,12], and micro-/nano-electromechanical devices[13–15]. Considering the 2D MXenes may undergo stretching, bending, and torsion in practical applications and result in the performance degradation[16,17], it is imperative to study the mechanical properties of MXenes.

To date, only a few theoretical and experimental studies have been conducted to investigate the mechanical properties of MXenes. Experimental studies on the mechanical properties of multilayer Ti$_3$C$_2$T$_x$ films of 40 nm thickness can reach 670 MPa as measured by in situ transmission electron microscopy (TEM) tensile tests[18]. However, these reported tensile strengths of multilayer Ti$_3$C$_2$T$_x$[19], which cannot reflect the true mechanical properties due to the weak interactions between monolayer 2D flakes, are significantly lower than the theoretical prediction of 20 GPa[20]. Therefore, the mechanical properties of monolayer Ti$_3$C$_2$T$_x$ nanosheets should be studied from the smallest component unit itself, which is the key in designing the structural stability and performance improvement of Ti$_3$C$_2$T$_x$-based materials.

Quantitative measurement of the mechanical properties of monolayer Ti$_3$C$_2$T$_x$ nanosheets is extremely challenging due to their nanoscale thickness[21,22]. Lipatov et al. conducted nanoindentation mechanical tests on monolayer Ti$_3$C$_2$T$_x$ by atomic force microscopy (AFM), and they reported an effective Young's modulus of 330 GPa

[1]Shanghai Key Laboratory of Intelligent Sensing and Detection Technology, East China University of Science and Technology, Shanghai 200237, P. R. China. [2]Key Laboratory of Pressure Systems and Safety of Ministry of Education, East China University of Science and Technology, Shanghai 200237, P. R. China. [3]School of Mechanical and Power Engineering, East China University of Science and Technology, Shanghai 200237, P. R. China. ✉e-mail: yanyabin@ecust.edu.cn; boweiz@ecust.edu.cn; fzxuan@ecust.edu.cn

(theoretically predicted value of 502 GPa)[23]. However, due to the limitation of the compression head tip size in the transverse local test area of $Ti_3C_2T_x$ nanosheets, highly inhomogeneous stress and strain fields are generated[24]. The different indenter positions as well as the internal stress existing in the samples will result in great uncertainty of the results[25]. Although the AFM nanoindentation method has been used to measure the mechanical properties of 2D materials such as graphene[26] and h-BN[27], these monolayer materials only have a single atomic layer, whereas the main body of monolayer $Ti_3C_2T_x$ has five atomic layers. Because the AFM method is perpendicular to the basal plane of 2D $Ti_3C_2T_x$, the atomic layer that contacting the AFM probe may deviates and slips from the normally aligned atomic structure, resulting in a serious mis-arrangement of the atoms, which will cause the inhomogeneous stress field. It is therefore hard to accurately measure the mechanical properties of monolayer $Ti_3C_2T_x$ nanosheets by the AFM nanoindentation method. Therefore, a reliable, direct, and quantitative method to measure the mechanical properties of monolayer $Ti_3C_2T_x$ nanosheets is urgently needed. Through the uniaxial tensile test, uniform loading can be carried out directly in the 2D material plane[28], which is also the most effective method to study the mechanical properties of $Ti_3C_2T_x$.

In this work, we prepared high-quality large-size monolayer $Ti_3C_2T_x$ nanosheets and fixed them to a nanomechanical test platform "Push-to-Pull" (PTP) for in situ tensile experiments using a precisely controlled focused ion beam (FIB) cutting technique, and an improved dry transfer technique. The Young's modulus and tensile strength of the monolayer $Ti_3C_2T_x$ nanosheet were measured. Meanwhile, we validated the experimental data by molecular dynamic simulation (MD) theoretical modeling calculation. Broadly speaking, this work provides an effective strategy for nanomechanical testing of other 2D materials produced by mechanical stripping and provides guidelines for the wide application of materials that requiring special mechanical properties such as $Ti_3C_2T_x$-based flexible electronic devices.

## Results

### Transfer of monolayer $Ti_3C_2T_x$ samples

The successful transfer of a monolayer $Ti_3C_2T_x$ nanosheet (Fig. 1a) to the PTP device is a critical step for in situ nanomechanical testing. For this purpose, we developed a unique dry transfer approach (Fig. 1b), which was modified from the previous method[19]. Specifically, the prepared monolayer $Ti_3C_2T_x$ suspension was dropped on a 400 mesh copper net (without carbon film) and vacuum dried (the synthesis procedure of monolayer $Ti_3C_2T_x$ suspension is detailed in Supplementary Fig. 1). The monolayer nanosheets was attached to the edges of the copper mesh, which greatly facilitates the subsequent transfer process (if the monolayer suspension is dried on a flat carrier, the nanosheets will be difficult to be transferred as the van der Waals force). Afterward, one side of the nanosheet was glued by electron beam-deposited Pt onto the mechanical probe, and the other three sides of the nanosheet were cut by Ga-focused ion beam (FIB) to move the nanosheet. The obtained nanosheet was transferred to the 2.5 μm stretch region in the middle of PTP microdevice. The $Ti_3C_2T_x$ nanosheets suspended on the nanomechanical device are almost transparent due to their monolayer nature. The manipulator and $Ti_3C_2T_x$ nanosheet were cut and separated by FIB.

### Characterization of monolayer $Ti_3C_2T_x$ samples

As shown in the SEM image (Fig. 2a), both ends of the monolayer $Ti_3C_2T_x$ nanosheet were fixed to the PTP nanomechanical device by electron beam-deposited Pt, and the nanosheet suspended above the gap was milling through FIB to the desired shape and size for tensile testing. During the test, a probe was used to exert a pushing force on the hemispherical indenter (indicated by the red arrow). The PTP microdevice converts the pushing force into plane tensile force on $Ti_3C_2T_x$ through the "push-to-pull" mechanism with a loading rate 10 nm/s. The load value can be calculated from the conversion value of

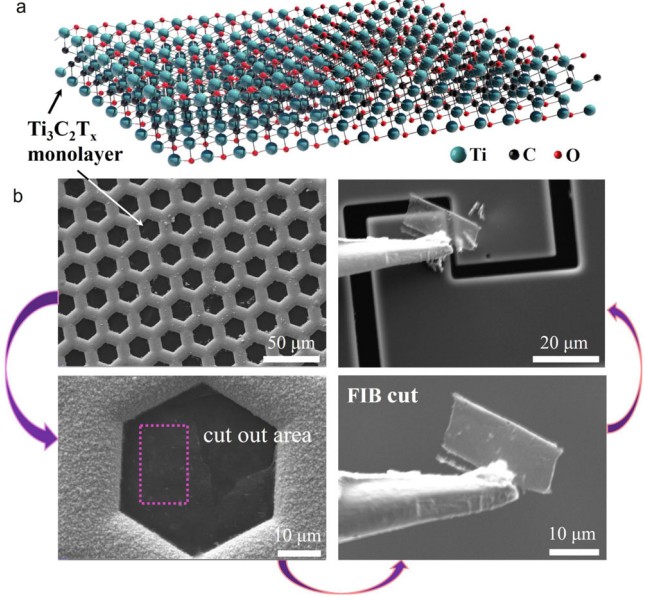

**Fig. 1 | Schematic and SEM images of monolayer $Ti_3C_2T_x$ transfer process. a** Structure of a $Ti_3C_2T_x$ monolayer. Note: the surface terminating groups $T_x$ shown can be several other parts, which can be fluoride (− F), oxy (= O), hydroxyl (− OH), and so on, only one surface group is shown in this figure. **b** Schematic illustration of $Ti_3C_2T_x$ monolayer transfer process. The sample carried on the edge of the copper mesh is transferred to the "Push-to-Pull" (PTP) nanomechanical device by FIB cutting and manipulator.

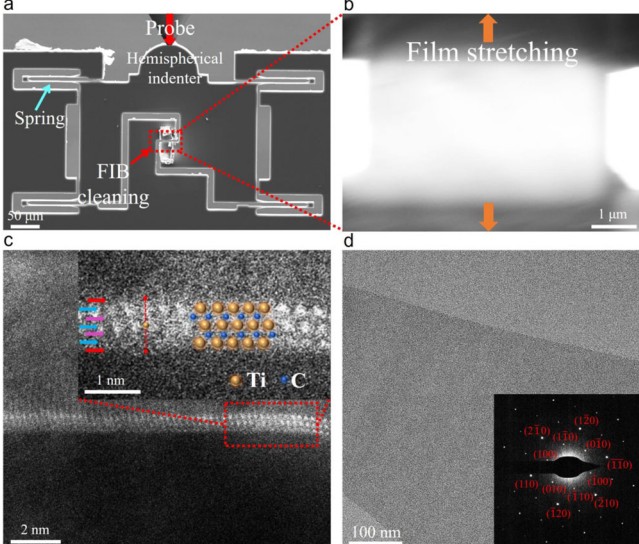

**Fig. 2 | Experimental steps and characterization of $Ti_3C_2T_x$ monolayer. a** The "Push-to-Pull" (PTP) nanomechanical device converts the compression on the hemispherical indenter into a tensile force on the sample under the SEM. **b** The enlarged view of the red area in (a) shows a $Ti_3C_2T_x$ monolayer of the stretched area that has been cut into a rectangle by FIB, the orange arrows represent the tensile direction. **c** Cross-sectional STEM image of $Ti_3C_2T_x$ monolayer observed along the fracture surface of the tested sample. Two C atomic layers (marked with purple arrows) are interwoven into three Ti-atomic layers (marked with blue arrows) in the order Ti(s)-C-Ti(c)-C-Ti(s) (Ti(s) stands for the Ti near the surface and Ti(c) for the central Ti), and the functional groups such as O and F atoms (marked with red arrows) are distributed on the surface of $Ti_3C_2$. **d** TEM image and crystalline SAED pattern of $Ti_3C_2T_x$ monolayer.

the electrostatic comb actuator in the planar probe, and the load-displacement data was recorded. Figure 2b shows the SEM image of a monolayer $Ti_3C_2T_x$ nanosheet after FIB milling. The width and length of $Ti_3C_2T_x$ are 5 µm and 2.5 µm, respectively, and the corresponding orange arrow direction is the sample tensile direction.

The thickness of the monolayer $Ti_3C_2T_x$ nanosheet is a critical parameter for the subsequent analysis of experimental results. Generally, the term thickness of monolayer 2D materials is ill-defined. For example, the thickness of monolayer graphene was measured using AFM with a value ranging from 0.4 to 1.7 nm[29]. However, in the actual study of calculating the mechanical properties, the nominal thickness of the monolayer graphene was used, which was 0.335 nm[26]. Nominal thickness has also been used in other studies to calibrate the thickness of 2D materials[30–32]. Although the thickness can be measured and estimated by AFM, the accuracy of this method is affected by different factors[29], such as the surface properties of $Ti_3C_2T_x$ and the interaction between the AFM tip and the surface of $Ti_3C_2T_x$. Similarly, the thickness of the monolayer determined by X-ray diffraction (XRD) depends on the water and other molecules embedded during the measurement[33]. Both methods can overestimate the nominal thickness of monolayer $Ti_3C_2T_x$, leading to uncertainty of measurement results. Therefore, in this work, the nominal thickness of monolayer $Ti_3C_2T_x$ of 0.98 nm was used[23]. As shown in Fig. 2c, the fracture edge cross-section of suspended $Ti_3C_2T_x$ nanosheet was characterized after a mechanical test using aberration-corrected scanning transmission electron microscopy (AC-STEM), which verified the thickness of monolayer $Ti_3C_2T_x$[34,35].

The properties of $Ti_3C_2T_x$ were confirmed by XRD (Supplementary Fig. 2a), X-ray photoelectron spectroscopy (XPS, Supplementary Fig. 2b–d, Supplementary Fig. 3), energy dispersion X-ray spectroscopy (EDX, Supplementary Fig. 4) and element mapping (Supplementary Fig. 5). The suspended $Ti_3C_2T_x$ nanosheets were also characterized by transmission electron microscopy (TEM). As shown in Fig. 2d, corresponding selected area electron diffraction (SAED) patterns with only one set of hexagonal diffraction patterns confirm the crystal nature and hexagonal carbide structure of $Ti_3C_2T_x$, showing the high quality of the $Ti_3C_2T_x$ nanosheets. During the Pt deposition process, Pt will inevitably propagate into the SEM chamber. Nevertheless, it is well known that the sprayed discontinuous Pt particles exhibit a loose and soft character, while $Ti_3C_2T_x$ nanosheets have strong and brittle mechanical properties, which won't exert substantial effects on the mechanical properties measurement. The crystal nature of the tested $Ti_3C_2T_x$ was confirmed by a series of SAED patterns from the edge to the center area in Supplementary Fig. 6a–c. Specifically, the fracture edge area of the tested $Ti_3C_2T_x$ sample on PTP device was further analyzed by TEM, as shown in Supplementary Fig. 6a, and the SAED pattern demonstrated that its crystal structure remained. In the edge area of the tested sample after being cut by FIB, the localized inhomogeneous was shown (Supplementary Fig. 6b) even though the minimum current was set and the corresponding SAED confirmed the unchanged crystal structure. For the center part of the tested $Ti_3C_2T_x$ MXene on PTP, Supplementary Fig. 6c shows no effects from $Ga^+$ sputtering and only a small amount of discontinuous Pt residue, and the corresponding SAED indicates a high quality. Supplementary Fig. 6d shows a low-magnification STEM image of the fracture edge area of the tested $Ti_3C_2T_x$ sample on the PTP, zoom-in view of the red rectangle area shows the fractured cross-sectional surface. Since the multilayer $Ti_3C_2T_x$ nanosheets are fractured layer by layer[18,36], but no multilayer structure and incomplete layers were observed from the fracture edges in Supplementary Fig. 6a and Supplementary Fig. 6c, d, indicating the monolayer nature of testing sample. Furthermore, the thickness of a large number of samples was measured by AFM. As shown in Supplementary Fig. 7, the percentage of monolayers exceeds 95% as shown in the thickness statistics (Supplementary Table 1), which

further substantiates the presence of monolayers in the resultant product.

## In situ tensile test of individual monolayer $Ti_3C_2T_x$ nanosheets

To investigate the elastic properties and tensile strength of monolayer $Ti_3C_2T_x$, the displacement-controlled in situ tensile experiments of monolayer $Ti_3C_2T_x$ were carried out in a field emission SEM. Prior to the tensile test, the hemispherical indenter of the PTP nanomechanical device was observed by SEM in the same plane as the mechanical probe. During the tensile process, both ends of the sample firmly adhered to the PTP device always, and no slippage was observed in the overlap area until fracture. The whole experiment was observed and recorded in real-time (as shown in Supplementary Movie 1, Supplementary Movie 2). To calculate the fracture strength of monolayer $Ti_3C_2T_x$, the maximum elongation of the sample needs to be measured during the stretching process.

As shown in Fig. 3a, b, through SEM snapshots before and after the tensile test, the maximum engineering strain of monolayer $Ti_3C_2T_x$ nanosheet before fracture can reach 3.6%. Figure 3c shows the sample completely failed with a typical brittle fracture. Figure 3d shows the corresponding load-displacement curve. It is worth noting that the slope of the curve was small in the initial stage, due to the mechanical probe just touching the hemispherical indenter, and the sample was not tensioned when only the spring of the nanomechanical device was driven. Therefore, the slope of the initial stage corresponds to the inherent stiffness of the nanomechanical device. The second stage was the process of the sample going from a relaxed state to being tightened. At the beginning of the orange arrow, the monolayer $Ti_3C_2T_x$ nanosheet was stretched. The slope of the curve in the third stage represents the total stiffness of the sample and the nanomechanical device. Starting from the purple asterisk, the sample was pulled off and the applied load drops sharply, and the probe continued to load until it stopped. The slope of the final stage is the same as the slope of the first stage, both of which are the inherent stiffness of the nanomechanical device. Here, the actual tensile stiffness of the monolayer $Ti_3C_2T_x$ nanosheet is equal to the total stiffness of the third stage minus the inherent stiffness of the nanomechanical device, and the actual tensile stiffness of the monolayer $Ti_3C_2T_x$ nanosheet can be calculated as ~947.7 N/m. The actual length and width of the measured sample region are 2.5 µm and 5 µm, and the 2D elastic modulus $E_{2D}$ = ~ 473.9 N/m is calculated. Under the continuum hypothesis[37,38], a Finite Element Method (FEM) model was established based on the experimental results to determine the uniaxial stress and strain of the specimen, and to study the distribution and evolution pattern of deformation and stress fields during loading (Supplementary Fig. 12). The results show that the stress hypothesis is more reasonable than the strain hypothesis for most areas of the sample. Therefore, under the uniaxial stress hypothesis, the 3D Young's modulus $E_{3D}$ = ~ 484 GPa is calculated by using $Ti_3C_2T_x$ nanosheet with a monolayer thickness of 0.98 nm.

Many monolayer $Ti_3C_2T_x$ nanosheets were used in our work. Unfortunately, due to the inherent difficulty of nanomechanical test operations and the fragility of monolayer $Ti_3C_2T_x$, most of them were failed to transfer. A total of five successful tests were conducted. Table 1 lists the dimensions, Young's modulus, and tensile strength of the tested samples. The average Young's modulus and fracture strength is 483.5 ± 13.2 GPa and 15.4 ± 1.92 GPa, respectively.

## Discussion

The effective Young's modulus ~484 GPa measured in the above experiments is close to the theoretically predicted value of 502 GPa by the molecular dynamics (MD) simulation[20]. Notably, Young's modulus of monolayer $Ti_3C_2T_x$ nanosheets measured using the PTP nanomechanical device far exceeds the previously reported value

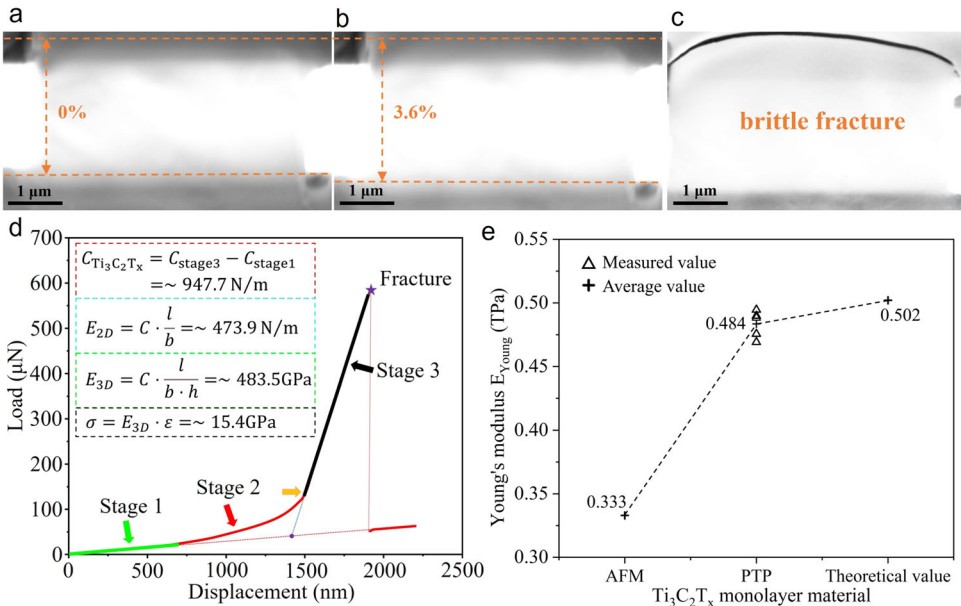

**Fig. 3 | Tensile fracture of monolayer Ti₃C₂Tₓ nanosheets and property comparison. a** SEM image shows that the Ti₃C₂Tₓ specimen was completely tightened at 0% strain. **b** SEM image of the sample before tensile fracture shows a peak strain of 3.6%. The orange dotted lines represent the two edges of the tested sample. **c** The brittle fracture morphology of the sample after failure, the associated results are listed in Table 1 (sample #4). **d** The measured load-displacement curve. The insertion formula shows the calculation process of mechanical properties (see section 5.2 for details). Tensile strength σ, 2D and 3D Young's modulus can be calculated, where $C$, $l$, $b$, $h$ and $\varepsilon$ are the tensile stiffness, the length in the stretched area, the width, the thickness, and the strain of the sample, respectively. **e** Comparison of Young's modulus of Ti₃C₂Tₓ monolayer from AFM indentation test, "Push-to-Pull" (PTP) in situ tensile and theoretical values.

(-330 GPa) by the nanoindentation method (Fig. 3e)[23], enabling a re-determination of the mechanical properties of monolayer Ti₃C₂Tₓ nanosheets. Compared with the mechanical properties of other monolayer 2D materials measured by in situ tensile experiments similar to PTP mechanisms (Supplementary Fig. 8), the effective Young's modulus of Ti₃C₂Tₓ MXene is higher than the average value of MoSe₂[39] but is lower than that of graphene[28]. Therefore, as promising candidates for micro-/nano-electromechanical devices that demand high mechanical qualities, as well as for serving as reinforcement materials in composites, Ti₃C₂Tₓ MXene presents a favorable alternative to graphene in the field of 2D materials. Noteworthily, both graphene and MoSe₂ are synthesized by chemical vapor deposition (CVD) techniques. Only SiO₂ and PMMA substrates need to be etched and removed, making the samples can be conveniently fixed with the test device directly and the measurement process simple. The Ti₃C₂Tₓ MXene nanosheets tested in this work were etched and stratified in solution due to the unique fabrication procedure, this significantly increases the difficulty of in situ mechanical testing of solution-treated samples. Furthermore, Ti₃C₂Tₓ MXene is only one of numerous 2D materials synthesized by solution-based techniques, the challenging step of sample transfer is instructive for the future investigation of the mechanical properties of solution-treated 2D materials. It is noteworthy that the effective Young's modulus of monolayer Ti₃C₂Tₓ nanosheets is three-orders-of-magnitude higher than the previously reported multilayer Ti₃C₂Tₓ[18]. This result substantiates the significance of monolayer measurements in order to unveil the intrinsic physical properties of MXene and its composites. Ti₃C₂Tₓ MXene has an effective engineering elastic strain of ~3.2% and exhibits brittle fracture, this would offer tremendous possibilities for MXene applications in strain engineering. In detail, the tensile strain leads to larger Ti-Ti bond lengths, and the internal stress will make the d-band center of Ti atoms closer to the Fermi energy level, giving Ti₃C₂Tₓ nanosheets abundant active sites, which will enhance the reactant/intermediate adsorption for accelerating the catalytic effect[40,41]. The elastic strain of ~3.2% is enough to give Ti₃C₂Tₓ materials excellent mechano-electrochemical coupling properties, which can be mechanically and chemically generated to change the structure of material to adjust their electronic structures and chemical properties, which also allows for numerous applications in the field of energy storage[42-44]. Applying strain and an electric field can also enable Ti₃C₂Tₓ MXene promising properties for optical nanodevices with a tunable band-gap electric field[45]. In addition, the elastic properties of Ti₃C₂Tₓ MXene make it suitable for applications in flexible robotic skin[46,47], structural composite films[48], protective coatings[49], and sensing fields[50].

The effective tensile fracture strength of ~15.4 GPa (Table 1) is lower than the theoretical value of ~18.4 GPa (Supplementary Table 2). To understand the origin of this difference, the morphology of monolayer Ti₃C₂Tₓ nanosheet was carefully in situ observed by the SEM/STEM, which didn't show any visible defects or cracks. We therefore believe that the decrease in tensile strength is cause by edge defects that are smaller than the characterization limit, which were formed during the FIB cutting and molding process, thereby reducing the fracture strength of monolayer Ti₃C₂Tₓ. To verify this point, we simulated the effect of edge defects on fracture strength by MD simulation. Based on recent studies of structural defects caused by ion radiation injection into 2D MXenes[51-53], three different types of edge defects were established and the width-scale dependency of the samples was demonstrated (Fig. 4a). The atomic structure of Ti₃C₂Tₓ is hexagonally arranged with inherent

### Table 1 | Mechanical properties of monolayer Ti₃C₂Tₓ nanosheets

| Sample # | Length (μm) | Width (μm) | Young's modulus E₃D (GPa) | Ultimate tensile strain (%) | Tensile strength (GPa) |
|---|---|---|---|---|---|
| 1 | 2.5 | 5 | 488.2 | 2.8 ± 0.196 | 13.7 ± 0.96 |
| 2 | 2.5 | 5 | 469.2 | 3.1 ± 0.217 | 14.5 ± 1.02 |
| 3 | 2.5 | 5 | 475.6 | 3.2 ± 0.224 | 15.2 ± 1.07 |
| 4 | 2.5 | 5 | 494.6 | 3.6 ± 0.252 | 17.8 ± 1.25 |
| 5 | 2.5 | 5 | 489.8 | 3.2 ± 0.224 | 15.7 ± 1.10 |
| Average | | | 483.5 ± 13.2 | 3.2 ± 0.652 | 15.4 ± 1.92 |

material orientations of armchair and zigzag shape[54]. The fracture strength of monolayer $Ti_3C_2T_x$ nanosheets of three different width scales was simulated along the two directions with both ends clamped respectively. The corresponding eighteen fracture strength results are shown in Fig. 4b. The vertical coordinate is the ratio of the ideal strength ($\sigma_0$) of the defect-free monolayer $Ti_3C_2T_x$ nanosheets to the fracture strength ($\sigma_m$) with implanted edge defects. The green shaded area shows the range of experimental values measured by the PTP method. From the simulated results, it can be seen that the edge defects induced during FIB cutting can indeed reduce their fracture strength. The experimental values of $\sigma_0/\sigma_m$ are in the range of 1.033 to 1.347. The simulated values of 18 types of different width scales and edge defects are close to the range of experimentally measured values. The effect of edge defects on the tensile strength of the sample diminishes with the increase of sample width, and the simulated values better fit the experimental values, indicating the effect of edge defects can be quantified. By employing a smaller FIB cutting current, the defect concentration at the edge of the sample can be effectively reduced

(Supplementary Fig. 9), thus improving the actual fracture strength of the $Ti_3C_2T_x$. Given the occurrence of Pt propagation and localized sputtering of $Ga^+$ during the experimental procedure, it is necessary to provide a comprehensive analysis and elucidation of the effects of Pt and $Ga^+$ on the mechanical test results of the samples. As previously mentioned, the Pt deposited onto the surface of monolayer $Ti_3C_2T_x$ nanosheets during testing has a soft nature and won't impact the mechanical properties of the strong and hard $Ti_3C_2T_x$ MXene. The results are additionally validated via experimental methods. Initially, high-energy Pt deposition is employed to fix both ends of $Ti_3C_2T_x$, resulting in the sample becoming opaque under SEM (Supplementary Fig. 10) due to Pt deposition on the sample surface, and the measured mechanical properties are shown in Table 1 (Sample #1–3). Subsequently, by setting the Pt deposition to low energy, the sample maintains the transparent nature, and the measured mechanical properties are presented in Table 1 (Sample #4–5). In addition, the SAED pattern of Supplementary Fig. 6a and Supplementary Fig. 6c indicate that the properties of the tested $Ti_3C_2T_x$ MXene are unchanged, and the comparison of experimental results confirms that the Pt deposition didn't exert a significant impact on the test. During the FIB cutting process, we set a minimum current of 1 pA to minimize the edge defect concentration. As can be seen from Supplementary Fig. 6a–c, the effect of $Ga^+$ on the samples was limited to the cut edges, and the crystal nature of the $Ti_3C_2T_x$ MXene was unchanged. It is worth noting that if there are a large number of defects in the 2D materials' internal region, which will obviously modulate the fracture behavior and result in multiple crack stages[55]. Furthermore, the fracture strength of the sample with edge defects remains in the same order of magnitude as the ideal strength of the defect-free sample, and the experimental measurements exceed half of the ideal value (i.e., deep ultra-strength[56]). These effectively demonstrate that the effect of FIB on the fracture strength of monolayer $Ti_3C_2T_x$ nanosheets is confined to the edge area only. In addition, this phenomenon has also been demonstrated in other studies using FIB to treat 2D materials[28,39,57]. Based on the above simulation results, the effect of edge defects on the tested samples has a width scale dependency. In this work, the width of the tested samples was deliberately fixed at 5 μm (maximize spanning the entire width of the tensile gap), which minimized the effect of edge defects on the tensile strength. Notably, this atomic edge defect is not a crack and has a negligible effect on the elastic modulus of the sample, which explains the results of our in situ tensile tests for individual monolayer $Ti_3C_2T_x$ nanosheets

In summary, we successfully realized the in situ mechanical stretching tests of individual monolayer $Ti_3C_2T_x$ nanosheets using the PTP nanomechanical device in SEM. Compared with the transverse localization test of the AFM nanoindentation test, the PTP device can achieve uniform stretching of the sample in the plane, and the mechanical properties of the monolayer $Ti_3C_2T_x$ can be reliably measured. The Young's modulus of monolayer $Ti_3C_2T_x$, 483.5 ± 13.2 GPa, is close to the theoretically predicted value of 502 GPa. The monolayer $Ti_3C_2T_x$ nanosheets exhibited brittle fracture with an average elastic strain of ~3.2%, which provides an opportunity for the application of $Ti_3C_2T_x$ in elastic strain engineering. Moreover, the difference between the experimentally effective fracture strength of 15.4 ± 1.92 GPa and the ideal value of ~18.4 GPa is attributed to the edge atomic-level defects of the sample, and this disparity diminishes as the width scale of sample increases. The effect of the edge defects on the fracture strength is quantified through molecular dynamic simulation, and the engineering fracture strength could be improved by modulating the edge state of monolayer $Ti_3C_2T_x$ nanosheets.

## Methods

### Sample preparation for tensile tests

The large-size monolayer $Ti_3C_2T_x$ nanosheets (Supplementary Fig. 1) were synthesized using an improved minimally intensive layer delamination (MILD). Specifically, we selected 200 mesh (74 μm) of MAX

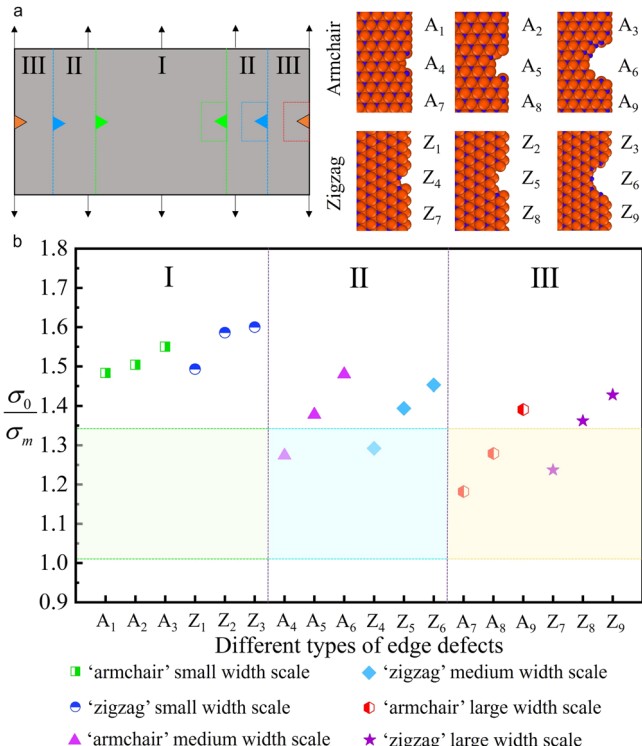

**Fig. 4 | MD simulations for the fracture strength of different width-scale $Ti_3C_2T_x$ monolayers with possible FIB-induced edged defects. a** MD simulation of $Ti_3C_2T_x$ nanoribbon tensile test. The colored triangles in the dashed box represent edge defects in the sample model. In the ball-and-stick model, the orange spheres represent Ti atoms and the blue spheres represent C atoms. The dash line box in the $Ti_3C_2T_x$ monolayers shows ball bar models of the atomic structures with representative edge defects, where I, II, III stand for the three width scales (I with a size of 122 × 95 Å, II with a size of 122 × 142 Å, III with a size of 122 × 190 Å), 'A' and 'Z' denote 'armchair' and 'zigzag', respectively. $A_1$, $A_4$, and $A_7$ represent small edge defect 'armchair' models, which increase in width size sequentially. $A_2$, $A_5$, and $A_8$ represent medium edge defect 'armchair' models, which increase in width size sequentially. $A_3$, $A_6$, and $A_9$ represent large edge defect 'armchair' models, which increase in width size sequentially. The serial number of the 'zigzag' model is the same. More specific model names are summarized in Supplementary Table 3. **b** The stress ratios obtained from the simulation results, where $\sigma_0$ is the ideal fracture strength of $Ti_3C_2T_x$ monolayer without defects, and $\sigma_m$ is the simulated value of $Ti_3C_2T_x$ monolayer with edge defects. Simulated data is marked with the corresponding width scale and defect name. The shaded areas represent the ratio of experimentally measured fracture strength versus the ideal fracture strength $\sigma_0$, the colors of the shaded areas correspond to the colors in (**a**).

phase $Ti_3AlC_2$, using the HCl + LiF etching method: 1.6 g LiF slowly dissolved in 40 mL of 6 mol/L HCl, stirring for 15 min to make it fully dissolved, and 1 g of $Ti_3AlC_2$ was slowly added. Etching at 35 °C for 30 h. The etched solution was first washed twice with HCl (1 M) to remove excess LiF, then washed and centrifuged with deionized water at a centrifugal speed of 684.8 × $g$ for 5 min each time, about 6-8 times, to make the solution pH greater than 6. The precipitate was collected after vacuum filtration and finally put into a vacuum drying oven and dried at 50 °C for 12 h. The large-size and high-quality monolayer $Ti_3C_2T_x$ suspension was obtained by the stratification method of manual shaking for 30 min and centrifugation at 684.8 × $g$ for 30 min.

## Sample transfer for tensile tests

A monolayer $Ti_3C_2T_x$ nanosheet was transferred to the PTP nanomechanical device by dry transfer technique for in situ tensile testing. The loss of sample size was cause by the Pt deposition and FIB cutting steps during the transfer process. We selected 400 mesh copper mesh (without carbon film) and pasted it directly on the SEM sample table of field emission SEM through conductive adhesive, and dropped monolayer suspension onto the copper mesh. After vacuum drying, the nanosheet was removed by a manipulator. However, the removed nanosheet may be bent due to the FIB and the manipulator, so a ring of Pt was deposited around the edge of the selected sample area for reinforcement before transfer. The above-removed nanosheet was aligned with the stretching area of the PTP device, and the sample cannot be broken at the moment of transfer to the PTP device. In the process of separating the manipulator from the $Ti_3C_2T_x$ nanosheet, FIB should also be used to cut completely to prevent the sample from being taken out of the original position. During the transfer of the nanosheets to the stretching area of PTP, the field emission SEM sample platform needs to be rotated 54°, and the PTP device is perpendicular to the camera of the FIB so that the cutting deviation will not occur during the cutting operation. Finally, the monolayer nanosheet was cut into shapes with a width of 5 μm and a length of 2.5 μm using FIB for the tensile test. In order to minimize irradiation damage during FIB cutting, an extra low accelerating voltage of 2 kV and small probe current of 1 pA were set.

## In situ SEM tensile testing

Monolayer $Ti_3C_2T_x$ nanosheets were tested in situ uniaxial tensile experiments in Carl Zeiss CossBeam340 SEM chamber, where the in situ tensile video was recorded at a low voltage of 5 kV to reduce electron beam effects. Before experiments, the Brukers-Hysitron PI88 picindenter (Supplementary Fig. 11a) was aligned with the semicircular indenter of the PTP device (Supplementary Fig. 11b), and the electrostatic comb driver parameters in the planar probe were calibrated by air. Under the displacement control of 10 nm/s, the PTP device was loaded using a PI88 probe, and the indenter sensor recorded the load-displacement curve. The PTP device converts the compression on the indenter into the uniaxial stretching of the sample. The tensile strain of the sample is measured directly from the SEM in situ tensile video, and the stress is calculated from the load divided by the cross-sectional area. By analyzing load-displacement curves, Young's modulus and tensile strength of monolayer $Ti_3C_2T_x$ nanosheets can be calculated. Specifically, the stiffness of the monolayer nanosheet is equal to the stiffness of the third stage of the load-displacement curve minus the stiffness of the nanomechanical device spring in the first stage. 2D and 3D Young's modulus are calculated as follows: $E_{2D} = C \bullet \frac{l}{b}$, $E_{3D} = C \bullet \frac{l}{b \bullet h}$, in the formula, $C$, $l$, $b$, and $h$ represent the tensile stiffness, length, width, and thickness of the sample in the tensile area respectively. Here, the length and width of the sample were 2.5 μm and 5 μm, respectively. In general, the term thickness is ill-defined for 2D materials, this work uses a "nominal" thickness for monolayer $Ti_3C_2T_x$ MXene, following a similar approach used for graphene. The thickness of the monolayer $Ti_3C_2T_x$ nanosheets was determined as 0.98 nm by

STEM and DFT. It is worth noting that the stress and strain states must be assumed because the sample is a monolayer nanosheet with nanometer-level thickness. In the actual tensile test, neck shrinkage did not occur on both transverse sides of the suspended sample. Combined with the simulation results of FEM analysis (Supplementary Fig. 12). The results show that we take any plane along the thickness direction, the stress state of the monolayer $Ti_3C_2T_x$ is the same, and the force in the thickness direction of the sample can be disregarded. In addition, the ultimate tensile strength of ~15.4 GPa can be calculated by considering the maximum tensile strain ε = ~3.2% as the formula of tensile strength $\sigma = E_{3D} \bullet \varepsilon$.

## Molecular dynamics (MD) simulation

Since it is almost impossible to observe the effect of edge defects on the mechanical properties of samples at the atomic level, the mechanical properties of monolayer $Ti_3C_2T_x$ nanosheets with edge atomic defects are investigated by MD simulation. All simulations were carried out in the Large-scale Atomic/Molecular Massively Parallel Simulator (LAMMPS). Uniaxial tensile modeling of monolayer $Ti_3C_2T_x$ was performed with the sample length of 122 Å, and width of 95 Å, 142 Å, 190 Å, respectively. The boundary conditions were periodic, along the armchair and zigzag directions, respectively. All three atomic defects were set at the edge of the sample. A uniform uniaxial tensile strain was applied at a loading rate of 0.05 $ps^{-1}$ before the atoms moved according to their equation of motion. Because the samples used in the experiment are high-quality single-crystal $Ti_3C_2T_x$ nanosheets, the rationality of atomic displacement is proved by assuming affine deformation of the crystal lattice. The temperature rise during loading is within 10 K. Therefore, the effect of thermal fluctuation on the simulation results is within a reasonable range, and the tensile deformation and fracture process of the sample is not affected. Using the third-generation COMB potential function COMB3 to describe the interactions between $Ti_3C_2T_x$ atoms, COMB3 is optimized for use with features that more realistically simulate interactions between different atoms to provide greater flexibility. The theoretically simulated fracture strength values ($\sigma_0$) for monolayer $Ti_3C_2T_x$ MXene nanosheets implanted with three edge defects of different widths are shown in Supplementary Table 3.

## Data availability

All data generated or analyzed during this study are included in the published article and its supplementary information files.

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

## Acknowledgements

B.Z. acknowledges the National Natural Science Foundation of China (Grant. No. 52105145, No. 12274124), the Shanghai Pilot Program for Basic Research (Grant. No. 22TQ1400100-6), and the Fundamental Research Funds for the Central Universities. F.-Z.X. thanks the Innovative Research Group Project of the National Natural Science Foundation of China (Grant. No. 52321002). Y.Y. acknowledges the National Natural Science Foundation of China (Grant. No. 52275149), and the Program for Professor of Special Appointment (Eastern Scholar) at Shanghai Institutions of Higher Learning.

## Author contributions

B.Z. conceived and planned the project. C.R., T.S., Z.L., and T.C. conducted the experiments and simulations. C.R., B.Z., and Y.Y. wrote the paper. All authors discussed the results and commented on the manuscripts.

## Competing interests

The authors declare no competing interests.
