## [Peer Review File · Nature Communications]

REVIEWER COMMENTS

Reviewer #1 (Remarks to the Author):

In this manuscript Rong et al. reported on experimental measurements of elastic properties of 2D $\text{Ti}_3\text{C}_2\text{T}_x$ monolayer. Using cutting edge technologies and PTP approach authors successfully performed in-situ measurements of elastic properties namely the effective Young's modulus, elastic stiffness and tensile strength with a higher robustness than previous experimental attempts. Additionally, authors directly compared experimental results with previous theoretical prediction and proposed them to be consistent. The most disputed part is related to performed MD calculations where reference values and new ones are mixed in unclear manner. Another weak point is very scarce discussion of practical importance of MXene application and its comparison with other materials in terms of elastic properties. Instead, authors focused on methodology primarily. The work could potentially be published in Nature Communications, but only after major revision.

Q1. In page 4 there is a typo (DM instead of MD).

Next, MD simulation part has to be carefully checked and clarified. As far as I understand from Fig. 4c, authors used value of $\sigma_{0,Ref} = 20 \text{ GPa}$ (from Ref. 20) to denote σ_{0}/σ_m region and show that obtained MD results are close enough to experimental ones.

Q2. Where particularly ideal strength theoretical value of "20 GPa" is presented in Ref. 20? If you used one from Figure 9b (Ti_3C_2), which particular strain rate was you looking at and why? In Ref. 20 values of 0.0002 ps^{-1} , 0.0004 ps^{-1} and 0.001 ps^{-1} rate are used, while in your methods section 0.05 ps^{-1} is provided. Also, rather different potential methods are used in Ref. 20 and this work. Please explain that.

Q3. In accordance with page 11, two green dashed lines in Fig. 4c are derived as $\sigma_{0,Ref}/\max(\text{experimental}) = 20 \text{ GPa} / 13.7 \text{ GPa} = 1.46$ and $\sigma_{0,Ref}/\min(\text{experimental}) = 20 \text{ GPa} / 13.1 \text{ GPa} = 1.53$. Here $\sigma_{0,Ref}$ is taken from Ref. 20.

Then authors performed own MD calculations (6 points in Fig. 4c). Please clarify which ideal strength $\sigma_{0,MD}$ value (I denote it in this manner to distinguish further) did you use for these

simulation points: again reference value ($\sigma_{0,Ref} = 20$ GPa) or your own $\sigma_{0,MD}$ value?

If numerator σ_0 is the same in both cases (i.e. $\sigma_{0,Ref} = \sigma_{0,MD} = 20$ GPa), the reasonable question is why you do not calculate σ_0 by yourselves? Also, if numerator is the same then in Fig. 4c you may directly compare $\sigma_{m,MD}$ and experimental $\sigma_{m,exp}$ values.

If numerator σ_0 is different, which particular values $\sigma_{0,MD}$ did you obtain from your MD? How do you compare $\sigma_{0,Ref}/\sigma_{m,exp}$ (green region) and $\sigma_{0,MD}/\sigma_{m,MD}$ (markers)? See Q2 again which refers to consistence requirement between Ref. 20 and your MD methods. Finally, why did you need σ_0 value from Ref. 20 at all?

From Fig. 4c caption I may propose that you calculated own $\sigma_{0,MD}$ values: «...The stress ratios obtained from the simulation results, where σ_0 is the ideal fracture strength of Ti_3C_2Tx monolayer without defects ...». In any case, please revise and explain this part carefully.

To summarize Q3: at least in Supplementary data please provide absolute values of σ_0 as well as σ_m for each of the six points in Fig. 4c.

Minor remark. In Fig. 4c usage of 6 different markers (shapes and colors) for 6 points is probably needless and may confuse a reader. Two types of markers (for ZZ and AC) seem to be enough.

Q4. Authors should better emphasize practical importance of the results even though the focus of the work is obviously kept on illustrative and high-quality experimental protocol. For instance, why «...exhibited brittle fracture with an elastic strain of ~2.8% provides an opportunity for the application of Ti_3C_2Tx in elastic strain engineering»? Only little discussion is provided while comparing graphene, $MoSe_2$ and MXene properties (page 10) without any real conclusions. Please extend this part.

Reviewer #2 (Remarks to the Author):

The authors have reported the mechanical properties monolayer MXene by means of push-to-pull devices (PTP) in tension. They reported a modulus of 478 GPa and strain to failure of 2.8%. The authors also studied the effect of edge defects on the strength of MXene via molecular dynamics simulations. The results show that the edge defects may reduce the load bearing capacity by ~40-60%, consistent with their experimental work. The paper shows a good combination of model and experiments.

However, the experimental part needs refinement, specially in relation to the uncertainty analysis. The authors have simply presented the standard deviation of the measurement as the uncertainty, whereas other sources of uncertainty such as strain measurement and thickness measurement are subjects of more uncertainty. In addition, the model can not be readily related to experiment due to potential length scale dependencies. More specific comments are below.

- I would like to see a more zoomed in image of Figure 2a. In particular, I am not clear what the gage length is? Is it taken to be the same as the free-standing length of $\sim 2.5 \text{ }\mu\text{m}$? For instance, consider the case where $2.5 \text{ }\mu\text{m}$ of the MXene is suspended, but on each side, $\sim 1 \text{ }\mu\text{m}$ overlaps with the device (is laid flat on the device), and then the it is FIBed. In this case, what is the actual gage length? Is it just $2.5 \text{ }\mu\text{m}$ or it is $2.5 + 1 + 1 = 4.5 \text{ }\mu\text{m}$?

With respect to the comment made above, the following statement requires clarification: “During the tensile process, both ends of the sample firmly adhered to the sample stage without significant slippage until fracture.” For instance, at a strain of $\sim 1\%$, a $2.5 \text{ }\mu\text{m}$ long MXene will elongate by only 25 nm . It is unclear to me how the authors can claim that there is no sliding?

- The other concern I have is the use of the thickness of 0.98 nm from Figure 2c. The uncertainty in thickness which may stem from that figure is large, and likely overshadow the uncertainty the authors claimed in the abstract. In general, the term thickness is ill-defined for 2D materials. I suggest that the authors use a “nominal” thickness for their MXene, the same way it is done for graphene.

- Another consideration is the measurement of the displacement from SEM images. Assuming that the image width is $\sim 5 \text{ }\mu\text{m}$ (estimated from Figure 3) with 1000 pixels, each pixel size corresponds to 5 nm , which is 0.2% ($\sim 7\%$ of the fracture strain). Hence, the displacement measurement via SEM by itself should add $\sim 7\%$ uncertainty which is $7\% * 478 \text{ GPa} = 33 \text{ GPa}$.

- One challenge with the simulation is that the edge effects will likely demonstrate length scale dependency (dependency of the fracture load on the width of the sample). The authors need to discuss that, and in light of this dependency relate the model to the experiment.

Reviewer #3 (Remarks to the Author):

The authors are trying to study the mechanical properties of Ti_3C_2 MXene. I agree with the authors that this topic requires more attention from the experimentalists, and all data published before requires further verification and confirmation.

I also think that the combination of a PTP device and in situ, SEM (or TEM) nanoindentation is probably the most suitable approach to studying the mechanical properties of the Ti_3C_2 monolayers.

So, the work is original (though the PTP technique was already used for tensile tests of MXene films <https://doi.org/10.1021/acsanm.1c00537>), and the results are new.

However, in the current version of the manuscript, several significant (and a couple of less important) issues need to be resolved/clarified before they can be considered for publication.

1. The authors claim that they used HAADF for thickness measurements. I see many problems with this approach. Fracture of MXene flakes is brittle and most probably will occur layer-by-layer (see <https://doi.org/10.1021/acs.nanolett.0c01861> figure 3d,e,f, also see attached scheme#1) and in this case, with HAADF we can only see the layered with was broken the last, not whole thickness of tested MXene flake.
2. The thickness of tested flakes needs to be confirmed. Authors can fix PTP with flake vertically in an SEM chamber (see scheme#2, attached) and measure the thickness of the tested flakes. For proper testing, this needs to be done before and after the tensile test.
3. Another comment about thickness, when authors talked about Figure 1b, they mentioned: "The Ti₃C₂T_x nanosheets suspended on the nanomechanical device are almost transparent due to their monolayer nature.". But in Figure 2b, Figure 3 and on videos (SI), flakes are not transparent. 1nm thick should be transparent in SEM even at 2kv (authors claim the test video was recorded at 5kV). Flakes on Figure 3 (a, b, c) do not look like 1nm thick.
4. In one of the attached videos (one 8 seconds long) of the tensile test, we can see a lot of redeposited material (probably Pt). This material is everywhere, including flake, which will affect the thickness and mechanical properties of tested MXene. EDS of flake on PTP device will be beneficial. Also, the flake on this video is 100% not monolayer (I have attached a screenshot from the video).
5. What were the uncertainties for thickness measurement? Authors need to consider that flake can be not fully parallel to the e-beam. This will cause an additional error in the measurements. I can even see from image 2c that the flake is more horizontal than vertical in the left part of the image.
6. Very likely, FIB milling may damage the MXene flake. Moreover, Pt may deposit on flake during transferring to the PTP device (see my comment#4). Thus, SAED diffraction of flakes on the PTP device must be done to confirm that the structure of MXene was not changed during the transferring process. (In addition, SAED after tests also may be very interesting and important)
7. Also, lower mag TEM and HAADF images of fracture need to be provided in addition to HR-HAADF. These images may confirm or disprove the authors' claim that this was monolayers. Also, if samples were indeed monolayers, authors can relatively easily get HAADF images not only for bent parts of the flakes but also for parts with an in-plane orientation (with 110 planes).
8. The thickness of as-synthesised MXene flakes also needs to be additionally confirmed by AFM (authors can apply their suspension to the silicon wafer and test a relatively large number of flakes (10-50) what % will be with a thickness less than 1nm? This is to confirm that there are monolayers in the final product.

9. There are not enough details about the indentation of empty PTP devices and information on the Elastic modulus of those. This data needs to be added to SI.

10. How were uncertainties for Young modulus and strength calculated? Clarification required.

11. In Figure S5, there are white dots on the HAADF image. What are those? Can these dots affect mechanical properties?

12. In section 2.1 authors claim: "For this purpose, we developed a unique dry transfer approach (Figure 1b)." This method is not unique in reference 19 (<https://doi.org/10.1021/acs.nanolett.0c01861>). Exactly the same transferring technique was used for in situ TEM tests of Ti₃C₂ MXene.

13. What FIB instrument was used for transferring and smiling the flake for a bone-like shape? Was it Ga or Xe FIB, and what conditions (and most importantly, Ion beam current) were used for milling the flake? Mxene monolayer will be extremely sensitive to ion beam. More details are required. Pt deposition was used for glueing the flakes to the manipulator and PTP device. Was it i-beam Pt deposition or e-beam Pt deposition?

To conclude, I think the methodology of the mechanical tests (particularly thickness measurements for taste MXene flakes) needs to be improved.

And In the current state, I cannot confirm that the measured values for Elastic modulus and strength are correct. More evidence is required.

Reviewer #4 (Remarks to the Author):

In this work, the authors measured the Young's modulus and tensile behaviour of monolayer Ti₃C₂T_x nanosheets using a push-to-pull in situ tensile testing platform. The experimental results were compared with theoretical calculations. The experimental Young's modulus is consistent with the theoretical value predicted by molecular dynamics simulation. However, the experimental tensile strength is much lower than the theoretical value. Through molecular dynamics simulation, the authors concluded that the lower tensile strength is attributed to the edge defects caused by FIB milling.

Ti₃C₂T_x is a 2D material with only 5 atomic layers. In general, 2D materials (e.g. graphene, MoS₂, etc.) are not very robust under high energy electron probe / ion beam. Many critical experimental parameters are lacking and some of the experimental procedures are not technically sounding (e.g. using FIB to process specimens with a thickness of few atomic layers). Also, the analysis of some of the experimental results are not very rigorous (see the comments below). Overall, the quality of the manuscript (scientifically and writing) does not meet the high standards of the journal.

1. The authors claimed that it is difficult to accurately measure the mechanical properties of monolayer Ti₃C₂T_x nanosheets by AFM nanoindentation as the indentation will cause dislocation in the crystal structure. Can you elaborate on why indentation will cause dislocation in the structure and how it can be avoided in a tensile test?

2. The thickness of the nanosheet was measured by STEM ADF image, as shown in Figure 2. What is the grey contrast above the atomic columns? Also, what are the parameters used for STEM imaging (accelerating voltage, probe current, etc)? Such information is critical, especially in the characterization of 2D materials which are sensitive to electron beam irradiation damage.

3. How was the elemental composition quantified in the EDX data set? Also, what is the source of fluorine? From the etchant?

4. In the HAADF image of Figure S5, there is some bright contrast spots. What is the origin of the bright contrast?

5. There are many unlabelled peaks in the X-ray photoelectron spectrum. Auger peaks should be labelled as well.

6. "As shown in Figure 2d, the monolayer nature property of the sample is further confirmed by the presence of a dark line at the edge only."

First, Figure 2d is not a HRTEM image, it is a TEM image (bright field, if objective aperture was inserted?). Second, it is not possible to confirm the monolayer nature of the specimen “by the presence of a dark line at the edge only.” The contrast at the edge is strongly affected by the defocus condition. For instance, overfocus condition will result in dark fringe at the specimen edge.

7. There are lots of contrast in the SEM image in Figure 2b. What is the origin of the contrast? Surface roughness, wrinkles? Since the authors claimed that the specimen thickness is about 0.98 nm, I was surprised by the contrast as such thin specimen should be quite electron transparent even at 5 kV. I was expecting to see the edge of the MEMS through the specimen, but from the image, it looks like the specimen is significantly thicker than 0.98 nm.

8. The experimental section does not contain sufficient FIB processing parameters. 2D materials are sensitive to ion beam induced damage. What were the accelerating voltage and probe current?

9. Pt deposition was used to manipulate and mount the sample. Is the deposition assisted by Ga⁺ ion beam or electron beam? The precursor gas is usually adsorbed / deposited in the area beyond the targeted region, potentially into the region of interest for the tensile test. If there is no subsequent thinning / cleaning of the specimen, did the authors consider the influence of contamination (potentially the bright contrast in the HAADF image Figure S5) on the tensile response of the specimen?

10. After mounting the monolayer nanosheet, it was cut into desired shape and geometry using FIB. During this process, the entire specimen was likely exposed to the ion beam (from imaging, positioning of the milling pattern, etc.). Since the specimen is a monolayer with a thickness of 5 atomic layers, the influence of the Ga⁺ ion beam on the specimen would be quite significant (specimen damage, Ga implantation). Would this be a contributing factor to the lower observed tensile strength compared to the theoretical value, in addition to the edge defects claimed by the authors?

11. The authors claimed that by using a lower FIB milling current, defect concentration at the edge of the sample can be reduced, resulting in higher fracture strength of Ti₃C₂T_x. However, there is no experimental evidence presented in the manuscript to support this claim.

Response to Reviewers

Reviewer #1

In this manuscript Rong et al. reported on experimental measurements of elastic properties of 2D $\text{Ti}_3\text{C}_2\text{T}_x$ monolayer. Using cutting edge technologies and PTP approach authors successfully performed in-situ measurements of elastic properties namely the effective Young's modulus, elastic stiffness and tensile strength with a higher robustness than previous experimental attempts. Additionally, authors directly compared experimental results with previous theoretical prediction and proposed them to be consistent. The most disputed part is related to performed MD calculations where reference values and new ones are mixed in unclear manner. Another weak point is very scarce discussion of practical importance of MXene application and its comparison with other materials in terms of elastic properties. Instead, authors focused on methodology primarily. The work could potentially be published in Nature Communications, but only after major revision.

Response: We are grateful for the reviewer's recommendation and helpful suggestions. The scientific nature of this paper was improved, especially on the MD calculations and discussion part, by absorbing the suggestions. Below are the addressed comments point by point:

Q1. In page 4 there is a typo (DM instead of MD).

Next, MD simulation part has to be carefully checked and clarified. As far as I understand from Fig. 4c, authors used value of $\sigma_{0, \text{Ref}} = 20$ GPa (from Ref. 20) to denote σ_0/σ_m region and show that obtained MD results are close enough to experimental ones.

Response: We thank the reviewer for the careful checking, and we have corrected this typo and checked the possible errors throughout the manuscript.

The MD simulation part has been revised based on the reviewer's suggestions, and the theoretical value $\sigma_{0, \text{Ref}}$ of the shadow part is taken using our own theoretical simulation $\sigma_{0, \text{MD}}$ value. Detail explanation is shown in the responses to questions Q2 and Q3.

“Meanwhile, we validated the experimental data by molecular dynamic simulation (MD) theoretical modeling calculation.”

Q2. Where particularly ideal strength theoretical value of “20 GPa” is presented in Ref. 20? If you used one from Figure 9b (Ti_3C_2), which particular strain rate was you looking at and why? In Ref. 20 values of 0.0002 ps^{-1} , 0.0004 ps^{-1} and 0.001 ps^{-1} rate are used, while in your methods section 0.05 ps^{-1} is provided. Also, rather different potential methods are used in Ref. 20 and this work. Please explain that.

Response: Thank you very much for your valuable comments. In the previous manuscript, we used Reference 20 as a standard for defect-free theoretical values. We used the ideal theoretical strength value for strain rate at 0.0002 ps^{-1} from **Figure 9b** in Reference 20. This is because the theoretical tensile strength at 0.0002 ps^{-1} is the closest to our experimental values. Therefore, in the shaded area, we did not use the theoretical strength of our simulated monolayer $\text{Ti}_3\text{C}_2\text{T}_x$ nanosheet, but instead made a direct reference to 20 GPa. At the points in **Figure 4** of this paper, our own simulated value for defect-free theoretical strength was used. However, the potential function in our simulations is different from that in Reference 20 as well as the selection of strain rate. For this part, we apologize for confusing reviewers with this unclear mix of reference and new values.

Therefore, according to the reviewer’s constructive suggestion in Q3, we used our own simulated theoretical tensile strength $\sigma_{0, \text{MD}}$ values for defect-free monolayer $\text{Ti}_3\text{C}_2\text{T}_x$ both in the shaded areas and at the points. The specific values were summarized in **Table S2** of Supplementary Information.

The strain rate in our simulations is 0.05 ps^{-1} , which is different from the strain rate in Ref. 20 due to the potential function being chosen differently. Different potential functions lead to different interatomic interactions, which correspond to different strain rates. Strain rates are examined in the calculations to find the optimized strain rate of 0.05 ps^{-1} (**Figure R1** in the response letter), where the dependency of the curves to the loading rate can be observed. With the rate of 0.1 ps^{-1} , the bonds break earlier due to a faster strain implementation. For the smaller engineering strain rates of 0.01 ps^{-1} , the stress-strain curve didn’t converge at a later stage. Therefore, the engineering strain rate of 0.05 ps^{-1} was selected to optimize the computational results.

Table S2. Theoretical simulated fracture strength values (σ_0) for three different widths of the defect-free monolayer $\text{Ti}_3\text{C}_2\text{T}_x$ MXene nanosheets in Figure 4.

Sample #	Length (Å)	Width (Å)	Fracture strength (GPa)
$\delta_{0-\text{I}}$	122	95	18.46
$\delta_{0-\text{II}}$	122	142	18.39
$\delta_{0-\text{III}}$	122	190	18.41

Figure R1. Stress-strain curve of $Ti_3C_2T_x$ MXene is presented using various strain rates with the size $122 \times 190 \text{ \AA}$.

Q3. In accordance with page 11, two green dashed lines in Fig. 4c are derived as $\sigma_{0, \text{Ref}}/\max(\text{experimental}) = 20 \text{ GPa}/13.7 \text{ GPa} = 1.46$ and $\sigma_{0, \text{Ref}}/\min(\text{experimental}) = 20 \text{ GPa}/13.1 \text{ GPa} = 1.53$. Here $\sigma_{0, \text{Ref}}$ is taken from Ref. 20.

Then authors performed own MD calculations (6 points in Fig. 4c). Please clarify which ideal strength $\sigma_{0, \text{MD}}$ value (I denote it in this manner to distinguish further) did you use for these simulation points: again reference value ($\sigma_{0, \text{Ref}} = 20 \text{ GPa}$) or your own $\sigma_{0, \text{MD}}$ value?

If numerator σ_0 is the same in both cases (i.e. $\sigma_{0, \text{Ref}} = \sigma_{0, \text{MD}} = 20 \text{ GPa}$), the reasonable question is why you do not calculate σ_0 by yourselves? Also, if numerator is the same then in Fig. 4c you may directly compare $\sigma_{m, \text{MD}}$ and experimental $\sigma_{m, \text{exp}}$ values.

If numerator σ_0 is different, which particular values $\sigma_{0, \text{MD}}$ did you obtain from your MD? How do you compare $\sigma_{0, \text{Ref}}/\sigma_{m, \text{exp}}$ (green region) and $\sigma_{0, \text{MD}}/\sigma_{m, \text{MD}}$ (markers)? See Q2 again which refers to consistence requirement between Ref. 20 and your MD methods. Finally, why did you need σ_0 value from Ref. 20 at all?

From Fig. 4c caption I may propose that you calculated own $\sigma_{0, \text{MD}}$ values: «...The stress ratios obtained from the simulation results, where σ_0 is the ideal fracture strength of $Ti_3C_2T_x$ monolayer without defects ...». In any case, please revise and explain this part carefully.

To summarize Q3: at least in Supplementary data please provide absolute values of σ_0 as well as σ_m for each of the six points in Fig. 4c.

Minor remark. In Fig. 4c usage of 6 different markers (shapes and colors) for 6 points is probably needless and may confuse a reader. Two types of markers (for ZZ and AC) seem to be enough.

Response: We appreciate the reviewer's insightful suggestions. In previous manuscripts, the numerator σ_0 is different, we used the defect-free theoretical strength from Reference 20 in the shaded areas and our own simulated defect-free theoretical strength at the points. However, the reference is distinct with our own modeled MD method. We are sorry for mixing our simulation values with the referenced simulation values that has confused the reviewer. Therefore, we took the valuable advice of the reviewer and used the defect-free theoretical $\sigma_{0, MD}$ value from our own simulations both in the shaded areas and at the points.

We use the σ_0/σ_m in the vertical coordinate in order to quantify the effect of edge defects on tensile strength in the next step and to make this change more intuitive. We also considered the dependency of the fracture strength on the width of the sample, and specific simulation data for 18 points were shown in **Table S3** of Supplementary Information. In **Figure 4b**, only two types of markers were used for presenting simulation results at the same width scale.

“...To verify this point, we simulated the effect of edge defects on fracture strength by MD simulation. Based on recent studies of structural defects caused by ion radiation injection into 2D MXenes⁵¹⁻⁵³, three different types of edge defects were established and the width-scale dependency of the samples was demonstrated (Figure 4a). The atomic structure of $Ti_3C_2T_x$ is hexagonally arranged with inherent material orientations of armchair and zigzag shape⁵⁴. The fracture strength of monolayer $Ti_3C_2T_x$ nanosheets of three different width scales was simulated along the two directions with both ends clamped respectively. The corresponding eighteen fracture strength results are shown in Figure 4b. The vertical coordinate is the ratio of the ideal strength (σ_0) of the defect-free monolayer $Ti_3C_2T_x$ nanosheets to the fracture strength (σ_m) with implanted edge defects. The green shaded area shows the range of experimental values measured by the PTP method. From the simulated results, it can be seen that the edge defects induced during FIB cutting can indeed reduce their fracture strength. The experimental values of σ_0/σ_m are in the range of 1.033 to 1.347. The simulated values of 18 types of different width scales and edge defects are close to the range of experimentally measured values. The effect of edge defects on the tensile strength of the sample diminishes with the increase of sample width, and the simulated values better fit the experimental values, indicating the effect of edge defects can be quantified.... Furthermore, the fracture strength of the sample with edge defects remains in the same order of magnitude as the ideal strength of the defect-free sample, and the experimental measurements exceed half of the ideal value (i.e., deep ultra-strength⁵⁶). These effectively demonstrate that the effect of FIB on the fracture strength of monolayer $Ti_3C_2T_x$ nanosheets is confined to the edge area only. In addition, this phenomenon has also been demonstrated in other

studies using FIB to treat 2D materials^{28,39,57}. Based on the above simulation results, the effect of edge defects on the tested samples has a width scale dependency.”

Table S3. The simulated theoretical fracture strength values (σ_0) for monolayer $Ti_3C_2T_x$ MXene nanosheets implanted with three edge defects of different widths in Figure 4.

Sample #	Length (Å)	Width (Å)	Orientation	Fracture strength (GPa)
A ₁	122	95	Armchair	12.42
A ₂	122	95	Armchair	12.24
A ₃	122	95	Armchair	11.88
Z ₁	122	95	Zigzag	12.33
Z ₂	122	95	Zigzag	11.61
Z ₃	122	95	Zigzag	11.52
A ₄	122	142	Armchair	14.45
A ₅	122	142	Armchair	13.36
A ₆	122	142	Armchair	12.44
Z ₄	122	142	Zigzag	14.25
Z ₅	122	142	Zigzag	13.21
Z ₆	122	142	Zigzag	12.67
A ₇	122	190	Armchair	15.58
A ₈	122	190	Armchair	14.39
A ₉	122	190	Armchair	13.24
Z ₇	122	190	Zigzag	14.88
Z ₈	122	190	Zigzag	13.52
Z ₉	122	190	Zigzag	12.89

Figure 4. MD simulations for the fracture strength of different width-scale $Ti_3C_2T_x$ monolayers with possible FIB-induced edged defects. (a) MD simulation of $Ti_3C_2T_x$ nanoribbon tensile test. The dash line box in the $Ti_3C_2T_x$ monolayers shows ball bar models of the atomic structures with representative edge defects, where I, II, III stand for the three width scales (I with a size of $122 \times 95 \text{ \AA}$, II with a size of $122 \times 142 \text{ \AA}$, III with a size of $122 \times 190 \text{ \AA}$), 'A' and 'Z' denote 'armchair' and 'zigzag', respectively. (b) The stress ratios obtained from the simulation results, where σ_0 is the ideal fracture strength of $Ti_3C_2T_x$ monolayer without defects, and σ_m is the simulated value of $Ti_3C_2T_x$ monolayer with edge defects. Simulated data is marked with the corresponding width scale and defect name. The shaded areas represent the ratio of experimentally measured fracture strength versus the ideal fracture strength σ_0 .

Q4. Authors should better emphasize practical importance of the results even though the focus of the work is obviously kept on illustrative and high-quality experimental protocol. For instance, why «...exhibited brittle fracture with an elastic strain of $\sim 2.8\%$ provides an opportunity for the application of $Ti_3C_2T_x$ in

elastic strain engineering»? Only little discussion is provided while comparing graphene, MoSe₂ and MXene properties (page 10) without any real conclusions. Please extend this part.

Response: We appreciate this precious suggestion and agree that the practical importance of the results should be discussed. The results measured in this work can not only reveal the intrinsic physical properties of MXene and its composites, but its elastic strain engineering also enables numerous applications for catalysis, micro-/nano-electromechanical devices, coatings, sensing, and structural composite films. In addition, we also provide more valuable discussion while comparing graphene, MoSe₂, and MXene properties. We added the discussion in the manuscript, as the details shown below:

“Compared with the mechanical properties of other monolayer 2D materials measured by in-situ tensile experiments similar to PTP mechanisms (Figure S8), the effective Young’s modulus of Ti₃C₂T_x MXene is higher than the average value of MoSe₂³⁹ but is lower than that of graphene²⁸. Therefore, as promising candidates for micro-/nano-electromechanical devices that demand high mechanical qualities, as well as for serving as reinforcement materials in composites, Ti₃C₂T_x MXene presents a favorable alternative to graphene in the field of 2D materials. Noteworthy, both graphene and MoSe₂ are synthesized by chemical vapor deposition (CVD) techniques. Only SiO₂ and PMMA substrates need to be etched and removed, making the samples can be conveniently fixed with the test device directly and the measurement process simple. The Ti₃C₂T_x MXene nanosheets tested in this work were etched and stratified in solution due to the unique fabrication procedure, this significantly increases the difficulty of in-situ mechanical testing of solution-treated samples. Furthermore, Ti₃C₂T_x MXene is only one of numerous 2D materials synthesized by solution-based techniques, the challenging step of sample transfer is instructive for the future investigation of the mechanical properties of solution-treated 2D materials. It is noteworthy that the effective Young’s modulus of monolayer Ti₃C₂T_x nanosheets is three-orders-of-magnitude higher than the previously reported multilayer Ti₃C₂T_x¹⁸. This result substantiates the significance of monolayer measurements in order to unveil the intrinsic physical properties of MXene and its composites. Ti₃C₂T_x MXene has an effective engineering elastic strain of ~3.2% and exhibits brittle fracture, this would offer tremendous possibilities for MXene applications in strain engineering. In detail, the tensile strain leads to larger Ti-Ti bond lengths, and the internal stress will make the d-band center of Ti atoms closer to the Fermi energy level, giving Ti₃C₂T_x nanosheets abundant active sites, which will enhance the reactant/intermediate adsorption for accelerating the catalytic effect^{40,41}. The elastic strain of ~3.2% is enough to give Ti₃C₂T_x materials excellent mechano-electrochemical coupling properties, which can be mechanically and chemically generated to change the structure of material to adjust their electronic structures

and chemical properties, which also allows for numerous applications in the field of energy storage⁴²⁻⁴⁴. Applying strain and an electric field can also enable $Ti_3C_2T_x$ MXene promising properties for optical nanodevices with a tunable band-gap electric field⁴⁵. In addition, the elastic properties of $Ti_3C_2T_x$ MXene make it suitable for applications in flexible robotic skin^{46,47}, structural composite films⁴⁸, protective coatings⁴⁹, and sensing fields⁵⁰.”

Reviewer #2

The authors have reported the mechanical properties monolayer MXene by means of push-to-pull devices (PTP) in tension. They reported a modulus of 478 GPa and strain to failure of 2.8%. The authors also studied the effect of edge defects on the strength of MXene via molecular dynamics simulations. The results show that the edge defects may reduce the load bearing capacity by ~40-60%, consistent with their experimental work. The paper shows a good combination of model and experiments. However, the experimental part needs refinement, specially in relation to the uncertainty analysis. The authors have simply presented the standard deviation of the measurement as the uncertainty, whereas other sources of uncertainty such as strain measurement and thickness measurement are subjects of more uncertainty. In addition, the model can not be readily related to experiment due to potential length scale dependencies. More specific comments are below.

Response: We sincerely thank the reviewer for careful review and the constructive suggestions towards improving our manuscript. The point-by-point responses are shown below:

1. I would like to see a more zoomed in image of Figure 2a. In particular, I am not clear what the gage length is? Is it taken to be the same as the free-standing length of ~2.5 μm ? For instance, consider the case where 2.5 μm of the MXene is suspended, but on each side, ~1 μm overlaps with the device (is laid flat on the device), and then the it is FIBed. In this case, what is the actual gage length? Is it just 2.5 μm or it is $2.5 + 1 + 1 = 4.5 \mu\text{m}$?

With respect to the comment made above, the following statement requires clarification: "During the tensile process, both ends of the sample firmly adhered to the sample stage without significant slippage until fracture." For instance, at a strain of ~1%, a 2.5 μm long MXene will elongate by only 25 nm. It is unclear to me how the authors can claim that the is no sliding?

Response: We thank the reviewer for the comments. The zoomed-in image of **Figure 2a** is shown in **Figure R2a**, where the actual gage length of the sample is 27 μm , the length of the sample suspended in the PTP test area is 2.5 μm , and the length of the sample overlapping the device is 24.5 μm . In **Figure R2b**, the actual gage length of another tested sample is 6.5 μm , and the length of the test area is 2.5 μm . The actual gage length of the tested sample depends on FIB cutting and transfer steps. The transverse dimensions of the sample need to be greater than 5 μm which ensures the set up even if there is a loss during the transfer process, and the length and width of sample in the stretched area are 2.5 μm and 5 μm , respectively. The nanosheet overlapped with the device is also conducive to the subsequent fixation step of Pt spray.

We are sorry for confusing the reviewer by this vague expression. In fact, we would like to express that the sample was firmly fixed onto the orange circled and rectangular areas (areas 1- 4) of PTP devices by Pt spray fixation, as shown in **Figure R2**. During the tensile test, the overlapped areas between the sample and the PTP device didn't slip. Therefore, we revised the expression of this sentence as follows:

“During the tensile process, both ends of the sample firmly adhered to the PTP device always, and no slippage was observed in the overlap area until fracture.”

Figure R2. *Ti₃C₂T_x MXene monolayer is fixed on the PTP nanomechanical device under the SEM. (a) The zoomed in image of Figure 2a. (b) The zoomed-in image of sample # 4 in Table 1. The orange parts in the figure represent the fixed areas where the Pt was sprayed.*

2. The other concern I have is the use of the thickness of 0.98 nm from Figure 2c. The uncertainty in thickness which may stem from that figure is large, and likely overshadow the uncertainty the authors claimed in the abstract. In general, the term thickness is ill-defined for 2D materials. I suggest that the authors use a “nominal” thickness for their MXene, the same way it is done for graphene.

Response: We appreciate this precious suggestion by the reviewer, which has been very helpful in improving the manuscript. We use a “nominal” thickness of 0.98 nm for the monolayer $Ti_3C_2T_x$ MXene in the manuscript, and **Figure 2c** is intended to demonstrate the monolayer nature of tested samples. This means the uncertainty of the monolayer $Ti_3C_2T_x$ thickness has a negligible effect on the measurement results.

*“The thickness of the monolayer $Ti_3C_2T_x$ nanosheet is a critical parameter for the subsequent analysis of experimental results. Generally, the term thickness of monolayer 2D materials is ill-defined. For example, the thickness of monolayer graphene was measured using AFM with a value ranging from 0.4 to 1.7 nm²⁹. However, in the actual study of calculating the mechanical properties, the nominal thickness of the monolayer graphene was used, which was 0.335 nm²⁶. Nominal thickness has also been used in other studies to calibrate the thickness of 2D materials³⁰⁻³². Although the thickness can be measured and estimated by AFM, the accuracy of this method is affected by different factors²⁹, such as the surface properties of $Ti_3C_2T_x$ and the interaction between the AFM tip and the surface of $Ti_3C_2T_x$. Similarly, the thickness of the monolayer determined by X-ray diffraction (XRD) depends on the water and other molecules embedded during the measurement³³. Both methods can overestimate the nominal thickness of monolayer $Ti_3C_2T_x$, leading to uncertainty of measurement results. Therefore, in this work, the nominal thickness of monolayer $Ti_3C_2T_x$ of 0.98 nm was used²³. As shown in **Figure 2c**, the fracture edge cross-section of suspended $Ti_3C_2T_x$ nanosheet was characterized after a mechanical test using aberration-corrected scanning transmission electron microscopy (AC-STEM), which verified the thickness of monolayer $Ti_3C_2T_x$ ^{34,35}”*

3. Another consideration is the measurement of the displacement from SEM images. Assuming that the image width is ~5 μ m (estimated from Figure 3) with 1000 pixels, each pixel size corresponds to 5 nm, which is 0.2% (~7% of the fracture strain). Hence, the displacement measurement via SEM by itself should add ~7% uncertainty which is 7%*478 GPa = 33 GPa.

Response: We thank the reviewer for pointing out this issue. We have added a 7% uncertainty to the ultimate tensile strain of the displacement measurement from SEM images in **Table 1** and recalculated the uncertainty of the measured results. The uncertainty of Young’s modulus E_{3D} is 13.2, and the uncertainty of Tensile strength σ is 1.92.

Uncertainty calculation of Young’s modulus E_{3D} :

$$\bar{E}_{3D} = 483.48, S_{E_{3D}} = \sqrt{\frac{\sum (E_{3D-i} - \bar{E}_{3D})^2}{n-1}} = 10.6288, \Delta_{E_{3D}} = \frac{t}{\sqrt{n}} \cdot S_{E_{3D}} \approx 13.2, \text{ where}$$

$n=5$, $\frac{t}{\sqrt{n}} = 1.24$. Therefore, the uncertainty of Young's modulus E_{3D} is 13.2.

Uncertainty calculation of Tensile strength σ :

$$\bar{\sigma} = 15.38, \quad S_{\sigma} = \sqrt{\frac{\sum(\sigma_i - \bar{\sigma})^2}{n-1}} = 1.54822, \quad \Delta_{\sigma} = \frac{t}{\sqrt{n}} \cdot S_{\sigma} \approx 1.92, \quad \text{where } n=5,$$

$\frac{t}{\sqrt{n}} = 1.24$. Therefore, the uncertainty of Tensile strength σ is 1.92.

Table 1. Mechanical properties of monolayer $Ti_3C_2T_x$ nanosheets. The displacement measurements by SEM itself should add ~7% uncertainty, the uncertainties in the measured values of Young's modulus and tensile strength are 2.7% and 12.5% respectively, which are within reasonable limits.

Sample #	Length (μm)	Width (μm)	Young's modulus E_{3D} (GPa)	Ultimate tensile strain (%)	Tensile strength (GPa)
1	2.5	5	488.2	2.8 ± 0.196	13.7 ± 0.96
2	2.5	5	469.2	3.1 ± 0.217	14.5 ± 1.02
3	2.5	5	475.6	3.2 ± 0.224	15.2 ± 1.07
4	2.5	5	494.6	3.6 ± 0.252	17.8 ± 1.25
5	2.5	5	489.8	3.2 ± 0.224	15.7 ± 1.10
Average			483.5 ± 13.2	3.2 ± 0.224	15.4 ± 1.92

4. One challenge with the simulation is that the edge effects will likely demonstrate length scale dependency (dependency of the fracture load on the width of the sample). The authors need to discuss that, and in light of this dependency relate the model to the experiment.

Response: We thank the reviewer for the constructive suggestion. We simulated the effect of different edge defects on the fracture strength of three width scale models, the sizes of the three models are $122 \times 95 \text{ \AA}$, $122 \times 142 \text{ \AA}$, and $122 \times 190 \text{ \AA}$ (length \times width), showing gradual increase in sample width. According to the simulation results, it is found that the larger the sample width, the smaller the

effect of edge defects on the fracture strength, and the closer the simulations to experimental values. In experimental tests, the size of the tensile area of the samples was $2.5 \mu\text{m} \times 5 \mu\text{m}$ (length \times width), which maximize the spanning width of tensile gap and minimize the effect of edge defects on the fracture strength of samples.

“...To verify this point, we simulated the effect of edge defects on fracture strength by MD simulation. Based on recent studies of structural defects caused by ion radiation injection into 2D MXenes^{51–53}, *three different types of edge defects were established and the width-scale dependency of the samples was demonstrated (Figure 4a)*. The atomic structure of $\text{Ti}_3\text{C}_2\text{T}_x$ is hexagonally arranged with inherent material orientations of armchair and zigzag shape⁵⁴. *The fracture strength of monolayer $\text{Ti}_3\text{C}_2\text{T}_x$ nanosheets of three different width scales was simulated along the two directions with both ends clamped respectively*. The corresponding *eighteen* fracture strength results are shown in **Figure 4b**. The vertical coordinate is the ratio of the ideal strength (σ_0) of the defect-free monolayer $\text{Ti}_3\text{C}_2\text{T}_x$ nanosheets to the fracture strength (σ_m) with implanted edge defects. The green shaded area shows the range of experimental values measured by the PTP method. From the simulated results, it can be seen that the edge defects induced during FIB cutting can indeed reduce their fracture strength. The experimental values of σ_0/σ_m are in the range of 1.033 to 1.347. The simulated values of 18 types of different width scales and edge defects are close to the range of experimentally measured values. The effect of edge defects on the tensile strength of the sample diminishes with the increase of sample width, and the simulated values better fit the experimental values, indicating the effect of edge defects can be quantified.... Furthermore, the fracture strength of the sample with edge defects remains in the same order of magnitude as the ideal strength of the defect-free sample, and the experimental measurements exceed half of the ideal value (i.e., deep ultra-strength⁵⁶). These effectively demonstrate that the effect of FIB on the fracture strength of monolayer $\text{Ti}_3\text{C}_2\text{T}_x$ nanosheets is confined to the edge area only. In addition, this phenomenon has also been demonstrated in other studies using FIB to treat 2D materials^{28,39,57}. Based on the above simulation results, the effect of edge defects on the tested samples has a width scale dependency. In this work, the width of the tested samples was deliberately fixed at $5 \mu\text{m}$ (maximize spanning the entire width of the tensile gap), which minimized the effect of edge defects on the tensile strength.”

Figure 4. MD simulations for the fracture strength of different width-scale $Ti_3C_2T_x$ monolayers with possible FIB-induced edged defects. (a) MD simulation of $Ti_3C_2T_x$ nanoribbon tensile test. The dash line box in the $Ti_3C_2T_x$ monolayers shows ball bar models of the atomic structures with representative edge defects, where I, II, III stand for the three width scales (I with a size of $122 \times 95 \text{ \AA}$, II with a size of $122 \times 142 \text{ \AA}$, III with a size of $122 \times 190 \text{ \AA}$), 'A' and 'Z' denote 'armchair' and 'zigzag', respectively. (b) The stress ratios obtained from the simulation results, where σ_0 is the ideal fracture strength of $Ti_3C_2T_x$ monolayer without defects, and σ_m is the simulated value of $Ti_3C_2T_x$ monolayer with edge defects. Simulated data is marked with the corresponding width scale and defect name. The shaded areas represent the ratio of experimentally measured fracture strength versus the ideal fracture strength σ_0 .

Reviewer #3

The authors are trying to study the mechanical properties of Ti_3C_2 MXene. I agree with the authors that this topic requires more attention from the experimentalists, and all data published before requires further verification and confirmation.

I also think that the combination of a PTP device and in situ, SEM (or TEM) nanoindentation is probably the most suitable approach to studying the mechanical properties of the Ti_3C_2 monolayers.

So, the work is original (though the PTP technique was already used for tensile tests of MXene films <https://doi.org/10.1021/acsnm.1c00537>), and the results are new.

However, in the current version of the manuscript, several significant (and a couple of less important) issues need to be resolved/clarified before they can be considered for publication.

Response: We thank the reviewer for the positive assessment and constructive suggestions of our work. The point-by-point responses are shown below:

1. The authors claim that they used HAADF for thickness measurements. I see many problems with this approach. Fracture of MXene flakes is brittle and most probably will occur layer-by-layer (see <https://doi.org/10.1021/acs.nanolett.0c01861> figure 3d, e, f, also see attached scheme#1) and in this case, with HAADF we can only see the layered with was broken the last, not whole thickness of tested MXene flake.

Response: We thank the reviewer for the valuable feedback. In general, the term thickness is ill-defined for 2D materials, we use a typical “nominal” thickness of 0.98 nm for the monolayer $\text{Ti}_3\text{C}_2\text{T}_x$ MXene in the manuscript. **Figure 2c** is intended to demonstrate the monolayer nature of our tested samples. For the case of MXene flakes probably will occur layer-by-layer fracture, we supplement the low-magnification TEM (**Figure R3**) and low-magnification STEM (**Figure R4**) of the fracture edges of the $\text{Ti}_3\text{C}_2\text{T}_x$ MXene on the PTP device, no layer-by-layer fracture is found at the fracture edges of samples after careful checking.

More importantly, due to the large number of defects and incomplete nanosheets exist in the thicker MXene nanosheets, the stress concentrated area often greatly reduces the experimental value of mechanical properties. However, the measured value of the nanosheets in this work is close to the theoretically predicted values of the monolayer $\text{Ti}_3\text{C}_2\text{T}_x$ MXene.

Moreover, we also demonstrated that the percentage of as-synthesized MXene monolayers exceeds 95%, as the AFM images shown in the following Figure S7.

Based on the above considerations, it is safe to conclude that our tested $\text{Ti}_3\text{C}_2\text{T}_x$ MXene is monolayer, and the “nominal” thickness is 0.98 nm.

Figure R3. TEM images of the fracture edges of monolayer $Ti_3C_2T_x$ MXene on a PTP device with corresponding SAED pattern.

Figure R4. STEM images of the fracture edges area of monolayer $Ti_3C_2T_x$ MXene on a PTP device. Zoom-in view of the red rectangle area shows the fractured cross-sectional surface of the tested sample.

“The thickness of the monolayer $Ti_3C_2T_x$ nanosheet is a critical parameter for the subsequent analysis of experimental results. Generally, the term thickness of monolayer 2D materials is ill-defined. For example, the thickness of monolayer graphene was measured using AFM with a value ranging from 0.4 to 1.7 nm²⁹. However, in the actual study of calculating the mechanical properties, the nominal thickness of the monolayer graphene was used, which was 0.335 nm²⁶. Nominal thickness has also been used in other studies to calibrate the thickness of 2D materials^{30–32}. Although the thickness can be measured and estimated by AFM, the accuracy of this method is affected by different factors²⁹, such as the surface properties of $Ti_3C_2T_x$ and the interaction between the AFM tip and the surface of $Ti_3C_2T_x$. Similarly, the thickness of the monolayer determined by X-ray diffraction (XRD) depends on the water and other molecules embedded during

*the measurement³³. Both methods can overestimate the nominal thickness of monolayer $Ti_3C_2T_x$, leading to uncertainty of measurement results. Therefore, in this work, the nominal thickness of monolayer $Ti_3C_2T_x$ of 0.98 nm was used²³. As shown in **Figure 2c**, the fracture edge cross-section of suspended $Ti_3C_2T_x$ nanosheet was characterized after a mechanical test using aberration-corrected scanning transmission electron microscopy (AC-STEM), which verified the thickness of monolayer $Ti_3C_2T_x$ ^{34,35}.”*

2. The thickness of tested flakes needs to be confirmed. Authors can fix PTP with flake vertically in an SEM chamber (see scheme#2, attached) and measure the thickness of the tested flakes. For proper testing, this needs to be done before and after the tensile test.

Response: Thank you very much for the suggestions. We have tried our best to fix the PTP with $Ti_3C_2T_x$ nanosheet vertically in the SEM chamber (**Figure R5**), but since the PTP device is located right in the center of the chip (**Figure R5a-b**), it is impossible to focus on and measure the thickness of the tested sample from the side of the PTP device. Therefore, we confirmed the monolayer nature of tested samples on the PTP device by low-magnification TEM and STEM at the fracture edges of the samples (layer-by-layer fracture didn't occur at the fracture edge of the sample), and selected the “nominal” thickness of 0.98 nm for the monolayer $Ti_3C_2T_x$ nanosheets (See response to Q1).

In addition, the thickness measurements by the AFM method also confirm that the proportion of monolayers in our prepared nanosheets is above 95% (**Figure S7**), and the average thickness is 1.17 nm (**Table S1**). Although the thickness can be measured and estimated by AFM, the accuracy of this method is affected by different factors, such as the AFM imaging mode (tapping, contact, etc.), tip-surface interactions, presence of various surface adsorbates, and trapped interfacial molecules (*Nanotechnology* 2016, 27, 125704; *ACS Nano* 2011, 5, 9703). Therefore, nominal thicknesses, rather than AFM-measured thicknesses, were used in other works on mechanical indentation of 2D materials for calculations of mechanical characteristics. Here, likewise, AFM produced a largely overestimated thickness value of 1.17 nm, and for Young's modulus calculation, we instead used the “nominal” thickness of a $Ti_3C_2T_x$ monolayer of 0.98 nm.

Figure R5. The PTP nanomechanical device under the SEM. (a-b) The PTP device on a chip. (c-d) The PTP device with flake is fixed vertically in the SEM chamber. (e) Tilt the PTP with the sample at an angle (The $\text{Ti}_3\text{C}_2\text{T}_x$ nanosheets suspended on the nanomechanical device are almost transparent due to their monolayer nature).

“Furthermore, the thickness of a large number of samples was measured by AFM. As shown in **Figure S7**, the percentage of monolayers exceeds 95% as shown in the thickness statistics (**Table S1**), which further substantiates the presence of monolayers in the resultant product.”

Figure S7. Thickness of $Ti_3C_2T_x$ MXene nanosheets. (a-d) AFM images of a large number of $Ti_3C_2T_x$ MXene nanosheets on the silicon substrate. The proportion of monolayer nanosheet is more than 95%. (e) Height profiles of MXene nanosheets in (a-d).

3. Another comment about thickness, when authors talked about Figure 1b, they mentioned: "The $Ti_3C_2T_x$ nanosheets suspended on the nanomechanical device are almost transparent due to their monolayer nature.". But in Figure 2b, Figure 3 and on videos (SI), flakes are not transparent. 1nm thick should be transparent in SEM even at 2kv (authors claim the test video was recorded at 5kv). Flakes on

Figure 3 (a, b, c) do not look like 1nm thick.

Response: We really appreciate this insightful question, which has been very helpful in improving the manuscript. In fact, in the previous step of fixing both ends of the sample, relatively abundant Pt was sprayed, resulting in the inevitable sputtering of Pt on the sample surface. The opacity of the sample in Figure 2b, Figure 3, and on videos (SI) is also due to the propagation of Pt. Therefore, we significantly reduced the sputtering of Pt by narrowing the spray area. The samples show the transparent nature of the monolayer in SEM (Figure 2b, Figure 3, and Video SI).

Figure 2. Experimental steps and characterization of $Ti_3C_2T_x$ monolayer. (a) The PTP nanomechanical device converts the compression on the hemispherical indenter into a tensile force on the sample under the SEM. (b) The enlarged view of the red area in (a) shows a $Ti_3C_2T_x$ monolayer of the stretched area that has been cut into a rectangle by FIB, the orange arrow represents the tensile direction. (c) Cross-sectional STEM image of $Ti_3C_2T_x$ monolayer observed along the fracture surface of the tested sample. Two C atomic layers (marked with purple arrows) are interwoven into three Ti-atomic layers (marked with blue arrows) in the order Ti(s)-C-Ti(c)-C-Ti(s), and the functional groups such as O and F atoms (marked with red arrows) are distributed on the surface of Ti_3C_2 . (d) TEM image and crystalline SAED pattern of $Ti_3C_2T_x$ monolayer.

Figure 3. Tensile fracture of monolayer $Ti_3C_2T_x$ nanosheets and property comparison. (a) SEM image shows that the $Ti_3C_2T_x$ specimen was completely tightened at 0% strain. (b) SEM image of the sample before tensile fracture shows a peak strain of 3.6%. (c) The brittle fracture morphology of the sample after failure, the associated results are listed in Table 1 (sample #4). (d) The measured load-displacement curve. The insertion formula shows the calculation process of mechanical properties (see section 5.2 for details). Tensile strength σ , 2D and 3D Young's modulus can be calculated, where C , l , b , h and ϵ are the tensile stiffness, the length in the stretched area, the width, the thickness, and the strain of the sample, respectively. (e) Comparison of Young's modulus of $Ti_3C_2T_x$ monolayer from AFM indentation test, PTP in situ tensile and theoretical values.

As for the effects of Pt deposition on the mechanical property of monolayer MXene, which was shown in the response for the following question (Q4).

4. In one of the attached videos (one 8 seconds long) of the tensile test, we can see a lot of redeposited material (probably Pt). This material is everywhere, including flake, which will affect the thickness and mechanical properties of tested MXene. EDS of flake on PTP device will be beneficial. Also, the flake on this video is 100% not monolayer (I have attached a screenshot from the video).

Response: We thank the reviewer for this feedback. Indeed, as the reviewer said, there is a lot of redeposited Pt on the surface of the tested $Ti_3C_2T_x$ nanosheet,

which is due to the fact that we sprayed relatively abundant Pt in the previous sample fixing step, resulting in the inevitable sputtering of Pt on the sample surface. Therefore, we significantly reduced the sputtering of Pt by narrowing the spray area and decreasing the amount of sprayed Pt, and the content of Pt is analyzed by EDS in the center area of the $\text{Ti}_3\text{C}_2\text{T}_x$ nanosheet on the PTP device (**Figure R6**). As can be seen from the TEM images of the center area of monolayer $\text{Ti}_3\text{C}_2\text{T}_x$ MXene on a PTP device, there are only a few discontinuous Pt particles on the sample surface (**Figure R7**).

Notably, it is widely acknowledged that the sprayed Pt particles exhibit a discontinuous and soft character (*Acta Materialia* 2012, 60, 2258; *Nano Letters* 2011, 11, 3207), while $\text{Ti}_3\text{C}_2\text{T}_x$ nanosheets have strong and brittle mechanical properties. Hence, the propagation of Pt didn't affect the mechanical properties of MXene. In addition, although the redeposition of Pt affects the thickness of the $\text{Ti}_3\text{C}_2\text{T}_x$ nanosheets, we are concerned about the thickness of monolayer $\text{Ti}_3\text{C}_2\text{T}_x$ nanosheets. The thickness of Pt was not considered in the thickness measurements of Figure S12a, b in this paper (<https://doi.org/10.1021/acs.nanolett.0c01861>), which measured only the thickness of MXene itself. Furthermore, we have experimentally verified that Pt deposition didn't affect the mechanical properties of monolayer $\text{Ti}_3\text{C}_2\text{T}_x$ nanosheets, evidenced from the data summarized in Table 1 below (Sample #4 and #5 have low Pt depositions showed similar results with the #1-#3 samples that have high Pt deposition).

Figure R6. EDS in the center area of the $\text{Ti}_3\text{C}_2\text{T}_x$ nanosheet on PTP device.

Figure R7. TEM images of the center area of monolayer $Ti_3C_2T_x$ MXene on a PTP device with corresponding SAED pattern.

“As previously mentioned, the Pt deposited onto the surface of monolayer $Ti_3C_2T_x$ nanosheets during testing has a soft nature and won't impact the mechanical properties of the strong and hard $Ti_3C_2T_x$ MXene. The results are additionally validated via experimental methods. Initially, high-energy Pt deposition is employed to fix both ends of $Ti_3C_2T_x$, resulting in the sample becoming opaque under SEM (**Figure S10**) due to Pt deposition on the sample surface, and the measured mechanical properties are shown in **Table 1** (Sample #1-3). Subsequently, by setting the Pt deposition to low energy, the sample maintains the transparent nature, and the measured mechanical properties are presented in **Table 1** (Sample #4-5).”

Table 1. Mechanical properties of monolayer $Ti_3C_2T_x$ nanosheets. The displacement measurements by SEM itself should add ~7% uncertainty, the uncertainties in the measured values of Young's modulus and tensile strength are 2.7% and 12.5% respectively, which are within reasonable limits.

Sample #	Length (μm)	Width (μm)	Young's modulus E_{3D} (GPa)	Ultimate tensile strain (%)	Tensile strength (GPa)
1	2.5	5	488.2	2.8 ± 0.196	13.7 ± 0.96
2	2.5	5	469.2	3.1 ± 0.217	14.5 ± 1.02
3	2.5	5	475.6	3.2 ± 0.224	15.2 ± 1.07
4	2.5	5	494.6	3.6 ± 0.252	17.8 ± 1.25

5	2.5	5	489.8	3.2 ± 0.224	15.7 ± 1.10
Average			483.5 ± 13.2	3.2 ± 0.652	15.4 ± 1.92

5. What were the uncertainties for thickness measurement? Authors need to consider that flake can be not fully parallel to the e-beam. This will cause an additional error in the measurements. I can even see from image 2c that the flake is more horizontal than vertical in the left part of the image.

Response: We thank the reviewer for this comment. We used the “nominal” thickness of a $\text{Ti}_3\text{C}_2\text{T}_x$ monolayer of 0.98 nm, which was determined by atomically resolved TEM and supported by theoretical calculations (see *J. Am. Chem. Soc.* 2015, 137, 2715–2721. *Chem. Mater.* 2014, 26, 2374–2381). **Figure 2c** is intended to demonstrate the monolayer nature of our tested samples, we supplement the low-magnification TEM (**Figure R3**) and low-magnification STEM (**Figure R4**) of the fracture edges of the $\text{Ti}_3\text{C}_2\text{T}_x$ MXene on the PTP device, no layer-by-layer fracture was found at the fracture edges of samples. Therefore, in **Figure 2c**, **Figure R3**, and **Figure R4**, we simply confirm the monolayer nature of the tested sample. The fact that the flake is not fully parallel to the electron beam would not affect our results as the fracture cross-section of the monolayer flake is clearly shown.

6. Very likely, FIB milling may damage the MXene flake. Moreover, Pt may deposit on flake during transferring to the PTP device (see my comment#4). Thus, SAED diffraction of flakes on the PTP device must be done to confirm that the structure of MXene was not changed during the transferring process. (In addition, SAED after tests also may be very interesting and important)

Response: We thank the reviewer for the constructive suggestion. The crystal nature of the tested $\text{Ti}_3\text{C}_2\text{T}_x$ can be confirmed by a series of SAED patterns from the edge to the center area in **Figure S6a-c**. The fracture edges area of the tested $\text{Ti}_3\text{C}_2\text{T}_x$ sample on the PTP device was further analyzed by TEM, as shown in **Figure S6a**, and the SAED pattern demonstrates that the crystal structure of the tested sample remained unchanged. The TEM in **Figure S6b** shows the edge area of the tested sample on PTP after being cut by FIB, and Ga^+ causes localized inhomogeneous sputtering at the sample edges even though the minimum current (1pA) has been set, corresponding SAED confirms that the crystal structure remains unchanged after FIB. **Figure S6c** shows a TEM image of the center area of the tested $\text{Ti}_3\text{C}_2\text{T}_x$ MXene on the PTP, which found no effects of Ga^+ sputtering

and only a small amount of discontinuous Pt residue, and the corresponding SAED indicates a high quality of the sample. Therefore, the effect of FIB on MXene flakes only causes edge defects, and the structure of the $\text{Ti}_3\text{C}_2\text{T}_x$ flakes on the PTP device was unchanged during the transferring process.

*“Given the occurrence of Pt propagation and localized sputtering of Ga^+ during the experimental procedure, it is necessary to provide a comprehensive analysis and elucidation of the effects of Pt and Ga^+ on the mechanical test results of the samples. As previously mentioned, the Pt deposited onto the surface of monolayer $\text{Ti}_3\text{C}_2\text{T}_x$ nanosheets during testing has a soft nature and won't impact the mechanical properties of the strong and hard $\text{Ti}_3\text{C}_2\text{T}_x$ MXene. The results are additionally validated via experimental methods. Initially, high-energy Pt deposition is employed to fix both ends of $\text{Ti}_3\text{C}_2\text{T}_x$, resulting in the sample becoming opaque under SEM (**Figure S10**) due to Pt deposition on the sample surface, and the measured mechanical properties are shown in **Table 1** (Sample #1-3). Subsequently, by setting the Pt deposition to low energy, the sample maintains the transparent nature, and the measured mechanical properties are presented in **Table 1** (Sample #4-5). In addition, the SAED pattern of **Figure S6a** and **S6c** indicate that the properties of the tested $\text{Ti}_3\text{C}_2\text{T}_x$ MXene are unchanged, and the comparison of experimental results confirms that the Pt deposition didn't exert a significant impact on the test. During the FIB cutting process, we set a minimum current of 1 pA to minimize the edge defect concentration. As can be seen from **Figure S6a-c**, the effect of Ga^+ on the samples was limited to the cut edges, and the crystal nature of the $\text{Ti}_3\text{C}_2\text{T}_x$ MXene was unchanged. It is worth noting that if there are a large number of defects in the 2D materials' internal region, which will obviously modulate the fracture behavior and result in multiple crack stages⁵⁵. Furthermore, the fracture strength of the sample with edge defects remains in the same order of magnitude as the ideal strength of the defect-free sample, and the experimental measurements exceed half of the ideal value (i.e., deep ultra-strength⁵⁶). These effectively demonstrate that the effect of FIB on the fracture strength of monolayer $\text{Ti}_3\text{C}_2\text{T}_x$ nanosheets is confined to the edge area only. In addition, this phenomenon has also been demonstrated in other studies using FIB to treat 2D materials^{28,39,57}.”*

Figure S6. TEM and STEM images of monolayer $Ti_3C_2T_x$ MXene on a PTP device. (a, b, c) The crystal structure of tested sample on the PTP device: Low-magnification TEM image of the fracture edges, the FIB-cut edges, the center area of $Ti_3C_2T_x$ MXene, and corresponding SAED pattern, respectively. (d) Low-magnification STEM image of the fracture edges area of $Ti_3C_2T_x$ MXene. Zoom-in view of the red rectangle area shows the fractured cross-sectional surface of the tested sample.

7. Also, lower mag TEM and HAADF images of fracture need to be provided in addition to HR-HAADF. These images may confirm or disprove the authors' claim that this was monolayers. Also, if samples were indeed monolayers, authors can relatively easily get HAADF images not only for bent parts of the flakes but also for parts with an in-plane orientation (with 110 planes).

Response: We thank reviewer very much for the helpful suggestion. Low-magnification TEM (**Figure R3**) and STEM (**Figure R4**) images of the fracture edges are provided, no layer-by-layer fracture was found at the fracture edges of

samples. Furthermore, the thickness of a large number of samples was measured by AFM, and the percentage of monolayers exceeds 95%, which further substantiates the presence of monolayers in the resultant product. The measured value of the monolayer $Ti_3C_2T_x$ MXene nanosheets in this work is close to the theoretically predicted values. Based on the above comprehensive analysis, it is safe to conclude that our tested $Ti_3C_2T_x$ MXene nanosheet is monolayer.

Figure R3. TEM images of the fracture edges of monolayer $Ti_3C_2T_x$ MXene on a PTP device with corresponding SAED pattern.

Figure R4. STEM images of the fracture edges area of monolayer $Ti_3C_2T_x$ MXene on a PTP device. Zoom-in view of the red rectangle area shows the fractured cross-sectional surface of the tested sample.

8. The thickness of as-synthesised MXene flakes also needs to be additionally confirmed by AFM (authors can apply their suspension to the silicon wafer and test a relatively large number of flakes (10-50) what % will be with a thickness less than 1nm? This is to confirm that there are monolayers in the final product.

Response: We really appreciate this insightful question. The thickness measurements by the AFM method also confirm that the proportion of monolayers in our prepared nanosheets is above 95% (**Figure S7**), and the average thickness is 1.17 nm (**Table S1**). Although the thickness can be measured and estimated by AFM, the accuracy of this method is often affected by different factors, such as the AFM imaging mode (tapping, contact, etc.), tip-surface interactions, presence of various surface adsorbates, and trapped interfacial molecules (*Nanotechnology* 2016, 27, 125704; *ACS Nano* 2011, 5, 9703). Therefore, nominal thicknesses, rather than AFM-measured thicknesses, were used in other works on mechanical indentation of 2D materials for calculations of mechanical characteristics. Here, likewise, AFM produced a largely overestimated thickness value of 1.17 nm, and for Young's modulus calculation, we instead used the "nominal" thickness of a $\text{Ti}_3\text{C}_2\text{T}_x$ monolayer of 0.98 nm.

*"The thickness of the monolayer $\text{Ti}_3\text{C}_2\text{T}_x$ nanosheet is a critical parameter for the subsequent analysis of experimental results. Generally, the term thickness of monolayer 2D materials is ill-defined. For example, the thickness of monolayer graphene was measured using AFM with a value ranging from 0.4 to 1.7 nm²⁹. However, in the actual study of calculating the mechanical properties, the nominal thickness of the monolayer graphene was used, which was 0.335 nm²⁶. Nominal thickness has also been used in other studies to calibrate the thickness of 2D materials³⁰⁻³². Although the thickness can be measured and estimated by AFM, the accuracy of this method is affected by different factors²⁹, such as the surface properties of $\text{Ti}_3\text{C}_2\text{T}_x$ and the interaction between the AFM tip and the surface of $\text{Ti}_3\text{C}_2\text{T}_x$. Similarly, the thickness of the monolayer determined by X-ray diffraction (XRD) depends on the water and other molecules embedded during the measurement³³. Both methods can overestimate the nominal thickness of monolayer $\text{Ti}_3\text{C}_2\text{T}_x$, leading to uncertainty of measurement results. Therefore, in this work, the nominal thickness of monolayer $\text{Ti}_3\text{C}_2\text{T}_x$ of 0.98 nm was used²³. As shown in **Figure 2c**, the fracture edge cross-section of suspended $\text{Ti}_3\text{C}_2\text{T}_x$ nanosheet was characterized after a mechanical test using aberration-corrected scanning transmission electron microscopy (AC-STEM), which verified the thickness of monolayer $\text{Ti}_3\text{C}_2\text{T}_x$ ^{34,35}."*

*"Furthermore, the thickness of a large number of samples was measured by AFM. As shown in **Figure S7**, the percentage of monolayers exceeds 95% as shown in the thickness statistics (**Table S1**), which further substantiates the presence of monolayers in the resultant product."*

Figure S7. Thickness of $Ti_3C_2T_x$ MXene nanosheets. (a-d) AFM images of a large number of $Ti_3C_2T_x$ MXene nanosheets on the silicon substrate. The proportion of monolayer nanosheet is more than 95%. (e) Height profiles of MXene nanosheets in (a-d).

Table S1. The thickness of monolayer $Ti_3C_2T_x$ MXene nanosheets in Figure S7 was statistically analyzed, and the average thickness was 1.17nm.

Sample #	Thicknesses (nm)	Sample #	Thicknesses (nm)	Sample #	Thicknesses (nm)

1	1.18	18	1.23	36	1.21
2	1.15	19	1.21	37	1.20
3	1.21	20	1.23	38	1.18
4	1.16	21	1.15	39	1.14
5	1.19	22	1.13	40	1.17
6	1.21	23	1.19	41	1.14
7	1.26	24	1.15	42	1.13
8	1.16	25	1.20	43	1.12
9	1.23	26	1.21	44	1.23
10	1.15	27	1.19	45	1.25
11	1.18	28	1.19	46	1.28
12	1.23	29	1.16	47	1.16
13	1.26	30	1.18	48	1.19
14	1.18	31	1.23	49	1.19
15	1.22	32	1.24	50	1.20
16	1.20	33	1.18	51	1.21
17	1.16	34	1.17	52	1.23
Average thicknesses (nm)				1.17	

9. There are not enough details about the indentation of empty PTP devices and information on the Elastic modulus of those. This data needs to be added to SI.

Response: We appreciate the reviewer’s suggestion and have added the stiffness and other details of the empty PTP device to the supplementary information:

“Push-to-Pull nanomechanical testing device

The theoretical stiffness of the empty PTP nanomechanical devices without

loading the sample is in the range of 20~100 N/m. The actual stiffness needs to be obtained from the slope of the force-displacement line at the initial stage during the test, or the slope of the tested sample after fracture. The probe is made of diamond and can apply a pushing force to the hemispherical indenter. The PTP device has four identical springs, symmetrically distributed at the corners so that the sample placed in the middle gap of the PTP device can be tensioned.”

Related pictures of the PTP devices are shown in the SI files.

10. How were uncertainties for Young modulus and strength calculated? Clarification required.

Response: Thank you very much for the comments. We have added a 7% uncertainty to the ultimate tensile strain of the displacement measurement from SEM images in **Table 1** and recalculated the uncertainty of the measured results. The uncertainty of Young’s modulus E_{3D} is 13.2, and the uncertainty of Tensile strength σ is 1.92.

Uncertainty calculation of Young’s modulus E_{3D} :

$$\bar{E}_{3D} = 483.48, S_{E_{3D}} = \sqrt{\frac{\sum (E_{3D-i} - \bar{E}_{3D})^2}{n-1}} = 10.6288, \Delta_{E_{3D}} = \frac{t}{\sqrt{n}} \cdot S_{E_{3D}} \approx 13.2, \text{ where}$$

$$n=5, \frac{t}{\sqrt{n}} = 1.24. \text{ Therefore, the uncertainty of Young’s modulus } E_{3D} \text{ is } 13.2.$$

Uncertainty calculation of Tensile strength σ :

$$\bar{\sigma} = 15.38, S_{\sigma} = \sqrt{\frac{\sum (\sigma_i - \bar{\sigma})^2}{n-1}} = 1.54822, \Delta_{\sigma} = \frac{t}{\sqrt{n}} \cdot S_{\sigma} \approx 1.92, \text{ where } n=5,$$

$$\frac{t}{\sqrt{n}} = 1.24. \text{ Therefore, the uncertainty of Tensile strength } \sigma \text{ is } 1.92.$$

Table 1. Mechanical properties of monolayer $Ti_3C_2T_x$ nanosheets. The displacement measurements via SEM by itself should add ~7% uncertainty, the uncertainties in the measured values of Young’s modulus and tensile strength are 2.7% and 12.5% respectively, which are within reasonable limits.

Sample #	Length (μm)	Width (μm)	Young’s modulus E_{3D} (GPa)	Ultimate tensile strain (%)	Tensile strength (GPa)
----------	--------------------------	-------------------------	--------------------------------	-----------------------------	------------------------

1	2.5	5	488.2	2.8 ± 0.196	13.7 ± 0.96
2	2.5	5	469.2	3.1 ± 0.217	14.5 ± 1.02
3	2.5	5	475.6	3.2 ± 0.224	15.2 ± 1.07
4	2.5	5	494.6	3.6 ± 0.252	17.8 ± 1.25
5	2.5	5	489.8	3.2 ± 0.224	15.7 ± 1.10
Average			483.5 ± 13.2	3.2 ± 0.224	15.4 ± 1.92

11. In Figure S5, there are white dots on the HAADF image. What are those? Can these dots affect mechanical properties?

Response: Many thanks for your careful observation. We are sorry that these white dots have confused the reviewer. In the previous manuscript, the nanosheets in **Figure S5** were irradiated under the STEM electron beam for too long, resulting in the structure damage of the MXene nanosheets (MXene is highly sensitive to the high-energy electron beam), leading to the occurrence of white dots. Therefore, we re-characterized the monolayer $Ti_3C_2T_x$ MXene nanosheet by HAADF with elemental mapping, as shown below.

Figure S5. HAADF and elemental mapping images of monolayer $Ti_3C_2T_x$ MXene nanosheet.

12. In section 2.1 authors claim: "For this purpose, we developed a unique dry transfer approach (Figure 1b)." This method is not unique in reference 19 (<https://doi.org/10.1021/acs.nanolett.0c01861>). Exactly the same transferring technique was used for in situ TEM tests of Ti_3C_2 MXene.

Response: Thanks much for the comments. Actually, we modified the methods based on reference 19 which was acknowledged and cited in the corresponding part of the manuscript.

The nanosheets in reference 19 are prepared on a TEM grid, the nanosheets and the substrate's TEM grid skeleton are cut down together by FIB (see **Figure S10** in <https://doi.org/10.1021/acs.nanolett.0c01861>). Whereas our samples are prepared directly on the edge of the carbon-free copper mesh and only the nanosheets were transferred during the transfer process. In addition, in reference 19, the sample was transferred and welded to the FIB grid. However, the transfer operation in our work requires the nanosheet to be transferred to the 2.5 μm stretch region of the PTP device, which is extremely challenging. Therefore, the transferring techniques are not exactly the same for both.

13. What FIB instrument was used for transferring and milling the flake for a bone-like shape? Was it Ga or Xe FIB, and what conditions (and most importantly, Ion beam current) were used for milling the flake? MXene monolayer will be extremely sensitive to ion beam. More details are required. Pt deposition was used for glueing the flakes to the manipulator and PTP device. Was it i-beam Pt deposition or e-beam Pt deposition?

Response: Thank you very much for your valuable comments. Ga^+ FIB was used for transferring and milling the flake for a bone-like shape. In order to minimize irradiation damage during FIB cutting, an extra low probe current of 1 pA was set. This is within a reasonable range because the FIB current for cutting monolayer 2D materials is generally set at 1 pA. The nanosheet and the mechanical probe were glued by e-beam Pt deposition. We have also added these details to the manuscript:

“Afterward, one side of the nanosheet was glued by electron beam -deposited Pt onto the mechanical probe, and the other three sides of the nanosheet were cut by Ga-focused ion beam (FIB) to move the nanosheet.”

*“As shown in the SEM image (**Figure 2a**), both ends of the monolayer $\text{Ti}_3\text{C}_2\text{T}_x$ nanosheet were fixed to the PTP nanomechanical device by **electron beam-deposited Pt**, and the nanosheet suspended above the gap was milling through FIB to the desired shape and size for tensile testing.”*

“In order to minimize irradiation damage during FIB cutting, an extra low accelerating voltage of 2 kV and small probe current of 1 pA were set”

Reviewer #4

In this work, the authors measured the Young's modulus and tensile behaviour of monolayer $\text{Ti}_3\text{C}_2\text{T}_x$ nanosheets using a push-to-pull in situ tensile testing platform. The experimental results were compared with theoretical calculations. The experimental Young's modulus is consistent with the theoretical value predicted by molecular dynamics simulation. However, the experimental tensile strength is much lower than the theoretical value. Through molecular dynamics simulation, the authors concluded that the lower tensile strength is attributed to the edge defects caused by FIB milling.

$\text{Ti}_3\text{C}_2\text{T}_x$ is a 2D material with only 5 atomic layers. In general, 2D materials (e.g. graphene, MoS_2 , etc.) are not very robust under high energy electron probe / ion beam. Many critical experimental parameters are lacking and some of the experimental procedures are not technically sounding (e.g. using FIB to process specimens with a thickness of few atomic layers). Also, the analysis of some of the experimental results are not very rigorous (see the comments below). Overall, the quality of the manuscript (scientifically and writing) does not meet the high standards of the journal.

Response: We appreciate the careful review and constructive suggestions on our manuscript. The scientific nature of this paper was improved by absorbing the suggestions. Below are addressed all the comments point by point:

1. The authors claimed that it is difficult to accurately measure the mechanical properties of monolayer $\text{Ti}_3\text{C}_2\text{T}_x$ nanosheets by AFM nanoindentation as the indentation will cause dislocation in the crystal structure. Can you elaborate on why indentation will cause dislocation in the structure and how it can be avoided in a tensile test?

Response: Thank you very much for your valuable comments. Because the AFM method is perpendicular to the direction of the $\text{Ti}_3\text{C}_2\text{T}_x$ basal plane, the atomic layer that first contacts the AFM probe would deviate and slip from the normally aligned atomic structure, resulting in a serious mis-arrangement of the atoms, which will cause dislocation in its crystal structure and an inhomogeneous stress field. The PTP nanodevice used in this work enables uniaxial uniform stretching of the $\text{Ti}_3\text{C}_2\text{T}_x$ nanosheets without causing inhomogeneous stress of the nanosheets.

“Although the AFM nanoindentation method has been used to measure the mechanical properties of 2D materials such as graphene²⁶ and h-BN²⁷, these monolayer materials only have a single atomic layer, whereas the main body of monolayer $\text{Ti}_3\text{C}_2\text{T}_x$ has five atomic layers. Because the AFM method is perpendicular to the basal plane of 2D $\text{Ti}_3\text{C}_2\text{T}_x$, the atomic layer that contacting

the AFM probe may deviates and slips from the normally aligned atomic structure, resulting in a serious mis-arrangement of the atoms, which will cause the inhomogeneous stress field. It is therefore hard to accurately measure the mechanical properties of monolayer $Ti_3C_2T_x$ nanosheets by the AFM nanoindentation method.”

2. The thickness of the nanosheet was measured by STEM ADF image, as shown in Figure 2. What is the grey contrast above the atomic columns? Also, what are the parameters used for STEM imaging (accelerating voltage, probe current, etc)? Such information is critical, especially in the characterization of 2D materials which are sensitive to electron beam irradiation damage.

Response: We are grateful to the reviewer for this valuable suggestion. In **Figure 2c**, the fracture edges of the $Ti_3C_2T_x$ MXene nanosheet was warped to a certain extent after the fracture of the sample, and the gray contrast above the atomic columns is the surface of the $Ti_3C_2T_x$ nanosheet.

The accelerating voltage used for STEM imaging is 300 kV. High angle annular dark field (HAADF)-STEM images were recorded using a convergence semi angle of 11 mrad, and inner- and outer collection angles of 59 and 200 mrad, respectively. In this work, STEM is used to characterize the fracture cross-section of $Ti_3C_2T_x$ nanosheets after testing, in order to prove the monolayer nature of the tested samples, so the effects of irradiation damage on the test results can be excluded. We have added relevant details in the supplementary information:

“Transmission electron microscopy (TEM) Characterization: Aberration-corrected STEM characterization was performed on a ThermoFisher Themis Z microscope equipped with two aberration correctors under 300 kV. High angle annular dark field (HAADF)-STEM images were recorded using a convergence semi angle of 11 mrad, and inner- and outer collection angles of 59 and 200 mrad, respectively.”

3. How was the elemental composition quantified in the EDX data set? Also, what is the source of fluorine? From the etchant?

Response: Thank you very much for your comments. The EDX spectrum was collected by spot-scanning the $Ti_3C_2T_x$ MXene film after vacuum filtration. This analysis shows the following chemical composition: Ti, 46 atom %; C, 23 atom %; O, 17 atom %; F, 12 atom %; and Cl, 2 atom %. Similar ratios between Ti, C, and surface termination groups were widely reported for other MXenes prepared via the MILD etching technique. The MXene synthesis method was widely reported and representative in current (*Sci. Adv.* 2018; 4: eaat049).

The fluorine is from the surface functional groups on the $Ti_3C_2T_x$ MXene, which

is originated from the chemical etching process by fluorine-containing solution. The general chemical formula $Ti_3C_2T_x$ MXene, T_x refers to the surface functional groups, which can be fluoride ($-F$), oxy ($=O$), or hydroxyl ($-OH$), exist on $Ti_3C_2T_x$ as a result of the topochemical conversion from Ti_3AlC_2 MAX phases to MXene, usually conducted in fluoride-containing aqueous solutions. The surface functional groups are the result of the chemical process that is used for the synthesis of MXene.

4. In the HAADF image of Figure S5, there is some bright contrast spots. What is the origin of the bright contrast?

Response: Many thanks for your careful observation. We are sorry that these white dots have confused the reviewer. In the previous manuscript, the nanosheets in **Figure S5** were irradiated under the STEM electron beam for too long, resulting in the structure damage of the MXene nanosheets (MXene is highly sensitive to the high-energy electron beam), leading to the occurrence of white dots. Therefore, we re-characterized the monolayer $Ti_3C_2T_x$ MXene nanosheet by HAADF with elemental mapping, as shown below.

Figure S5. HAADF and elemental mapping images of monolayer $Ti_3C_2T_x$ MXene nanosheet.

5. There are many unlabelled peaks in the X-ray photoelectron spectrum. Auger peaks should be labelled as well.

Response: We appreciate the reviewer's suggestion. The Auger peaks in the X-ray photoelectron spectrum are carefully labelled, as shown in the below images. The MXene shows similar chemical compositions with the reported cases (*Nano Letters* 2020, 20, 5900).

Figure S2b, c, d show high-resolution C 1s, Ti 2p, and O 1s XPS spectra of the $Ti_3C_2T_x$. The C 1s peak was fitted using four components located at 282.2, 284.6, 286.2, and 289.3 eV, corresponding to the C-Ti, C-C, C-O, and O-C=O bonds. The bindings of Ti $2p_{3/2}$ at 455.3, 456.0, 457.1, and 458.7 eV are attributed to the Ti-C, Ti^{+2} -C, Ti^{+3} -C bond, and Ti^{4+} ions ($TiO_{2-x}F_x$), respectively. Similarly, the main O 1s core-level peaks at 529.8, 530.3, 531.2, and 533.3 eV are related to TiO_2 , C-O, C-Ti-O, and C-Ti-OH in order

Figure S2. Characterization of the prepared monolayer $Ti_3C_2T_x$ MXene: (a) XRD pattern of $Ti_3C_2T_x$. High-resolution XPS spectra of the (b) C 1s, (c) Ti 2p, (d) O 1s from $Ti_3C_2T_x$.

6. “As shown in Figure 2d, the monolayer nature property of the sample is further confirmed by the presence of a dark line at the edge only.”

First, Figure 2d is not a HRTEM image, it is a TEM image (bright field, if objective aperture was inserted?). Second, it is not possible to confirm the monolayer nature of the specimen “by the presence of a dark line at the edge only.” The contrast at the edge is strongly affected by the defocus condition. For instance, overfocus condition will result in dark fringe at the specimen edge.

Response: We really appreciate this insightful question, which has been very helpful in improving the manuscript. As stated by the reviewer, **Figure 2d** is a

TEM image and cannot to confirm the monolayer nature of the specimen. The suspended $\text{Ti}_3\text{C}_2\text{T}_x$ nanosheet is characterized by TEM in **Figure 2d**, and corresponding selected area electron diffraction (SAED) patterns with only one set of hexagonal diffraction patterns confirm the hexagonal carbide structure of $\text{Ti}_3\text{C}_2\text{T}_x$, showing the high quality of the $\text{Ti}_3\text{C}_2\text{T}_x$ nanosheets.

Figure 2c is intended to demonstrate the monolayer nature of our tested samples. For the case of multi-layer MXene flakes probably will occur layer-by-layer fracture (see <https://doi.org/10.1021/acs.nanolett.0c01861> figure 3d, e, f), we supplement the low-magnification TEM (**Figure R3**) and low-magnification STEM (**Figure R4**) of the fracture edges of the $\text{Ti}_3\text{C}_2\text{T}_x$ MXene on the PTP device, no layer-by-layer fracture is found at the fracture edges of samples. In addition, due to the large number of defects and incomplete nanosheets in the thicker MXene nanosheets, the stress concentrated area often greatly reduces the experimental value of mechanical properties. However, the measured value of the nanosheets in this work is close to the theoretically predicted values of the monolayer $\text{Ti}_3\text{C}_2\text{T}_x$ MXene.

In addition, the thickness measurements by the AFM method also confirm that the proportion of monolayers in our prepared nanosheets is above 95% (**Figure S7**), and the average thickness is 1.17 nm (**Table S1**). Although the thickness can be measured and estimated by AFM, the accuracy of this method can be affected by different factors, such as the AFM imaging mode (tapping, contact, etc.), tip-surface interactions, presence of various surface adsorbates, and trapped interfacial molecules (*Nanotechnology* 2016, 27, 125704; *ACS Nano* 2011, 5, 9703). Therefore, nominal thicknesses, rather than AFM-measured thicknesses, were used in other works on mechanical indentation of 2D materials for the calculations of mechanical characteristics. Here, likewise, AFM produced an overestimated thickness value of 1.17 nm, and for Young's modulus calculation, we instead used the "nominal" thickness of a $\text{Ti}_3\text{C}_2\text{T}_x$ monolayer of 0.98 nm (*J. Am. Chem. Soc.* 2015, 137, 2715–2721. *Chem. Mater.* 2014, 26, 2374–2381). Therefore, our tested $\text{Ti}_3\text{C}_2\text{T}_x$ MXene is monolayer.

Figure R3. TEM images of the fracture edges of monolayer $Ti_3C_2T_x$ MXene on a PTP device with corresponding SAED pattern.

Figure R4. STEM images of the fracture edges area of monolayer $Ti_3C_2T_x$ MXene on a PTP device. Zoom-in view of the red rectangle area shows the fractured cross-sectional surface of the tested sample.

Figure S7. Thickness of $Ti_3C_2T_x$ MXene nanosheets. (a-d) AFM images of a large number of $Ti_3C_2T_x$ MXene nanosheets on the silicon substrate. The proportion of monolayer nanosheet is more than 95%. (e) Height profiles of MXene nanosheets in (a-d).

7. There are lots of contrast in the SEM image in Figure 2b. What is the origin of the contrast? Surface roughness, wrinkles? Since the authors claimed that the specimen thickness is about 0.98 nm, I was surprised by the contrast as such thin specimen should be quite electron transparent even at 5 kV. I was expecting to see the edge of the MEMS through the specimen, but from the image, it looks like the specimen is significantly thicker than 0.98 nm.

Response: We appreciate this comment. In fact, in the previous step of fixing both ends of the sample, relatively abundant Pt was sprayed on the nanosheet surface, resulting in the opaque nature under SEM. The inhomogeneous roughness of the tested sample surface is due to the sputtering of Pt. The opacity

of the sample in Figure 2b, Figure 3, and on videos (SI) was also caused by the Pt deposition. Therefore, we reduced the sputtering of Pt by narrowing the spray area and decreasing the amount of sprayed Pt. The samples show the transparent nature of the monolayer in SEM (Figure 2b, Figure 3, and Video SI), and the edges of the MEMS through the specimen became visible.

It is important to note that we have demonstrated the Pt deposition has no effect on the measurement of mechanical property of MXene monolayers from both the theoretical and experimental insights (see more detail from the manuscript).

Figure 2. Experimental steps and characterization of $Ti_3C_2T_x$ monolayer. (a) The PTP nanomechanical device converts the compression on the hemispherical indenter into a tensile force on the sample under the SEM. (b) The enlarged view of the red area in (a) shows a $Ti_3C_2T_x$ monolayer of the stretched area that has been cut into a rectangle by FIB, the orange arrow represents the tensile direction. (c) Cross-sectional STEM image of $Ti_3C_2T_x$ monolayer observed along the fracture surface of the tested sample. Two C atomic layers (marked with purple arrows) are interwoven into three Ti-atomic layers (marked with blue arrows) in the order Ti(s)-C-Ti(c)-C-Ti(s), and the functional groups such as O and F atoms (marked with red arrows) are distributed on the surface of Ti_3C_2 . (d) TEM image and crystalline SAED pattern of $Ti_3C_2T_x$ monolayer.

Figure 3. Tensile fracture of monolayer $Ti_3C_2T_x$ nanosheets and property comparison. (a) SEM image shows that the $Ti_3C_2T_x$ specimen was completely tightened at 0% strain. (b) SEM image of the sample before tensile fracture shows a peak strain of 3.6%. (c) The brittle fracture morphology of the sample after failure, the associated results are listed in Table 1 (sample #4). (d) The measured load-displacement curve. The insertion formula shows the calculation process of mechanical properties (see section 5.2 for details). Tensile strength σ , 2D and 3D Young's modulus can be calculated, where C , l , b , h and ϵ are the tensile stiffness, the length in the stretched area, the width, the thickness, and the strain of the sample, respectively. (e) Comparison of Young's modulus of $Ti_3C_2T_x$ monolayer from AFM indentation test, PTP in situ tensile and theoretical values.

8. The experimental section does not contain sufficient FIB processing parameters. 2D materials are sensitive to ion beam induced damage. What were the accelerating voltage and probe current?

Response: We thank the reviewer for this comment. Ga^+ FIB was used for transferring and milling the flake for a bone-like shape. In order to minimize irradiation damage during FIB cutting, an extra low accelerating voltage of 2 kV and small probe current of 1 pA were set. These are within the reasonable range, because the FIB current for cutting monolayer 2D materials is generally set at 1 pA and 2 kV. We have also added these details to the manuscript:

“In order to minimize irradiation damage during FIB cutting, an extra low accelerating voltage of 2 kV and small probe current of 1 pA were set.”

9. Pt deposition was used to manipulate and mount the sample. Is the deposition assisted by Ga⁺ ion beam or electron beam? The precursor gas is usually adsorbed / deposited in the area beyond the targeted region, potentially into the region of interest for the tensile test. If there is no subsequent thinning / cleaning of the specimen, did the authors consider the influence of contamination (potentially the bright contrast in the HAADF image Figure S5) on the tensile response of the specimen?

Response: We appreciate this precious suggestion. The electron beam Pt is used to manipulate and mount the sample. We have also added these details to the manuscript:

“Afterward, one side of the nanosheet was glued by electron beam -deposited Pt onto the mechanical probe, and the other three sides of the nanosheet were cut by Ga-focused ion beam (FIB) to move the nanosheet. ”

“As shown in the SEM image (Figure 2a), both ends of the monolayer Ti₃C₂T_x nanosheet were fixed to the PTP nanomechanical device by electron beam-deposited Pt, and the nanosheet suspended above the gap was milled through FIB to the desired shape and size for tensile testing.”

Indeed, there is a lot of redeposited Pt on the surface of the tested Ti₃C₂T_x nanosheet, which is inevitable from the sputtering fixation step. Therefore, we reduced the sputtering of Pt by narrowing the spray area and decreasing the amount of sprayed Pt. As can be seen from the TEM images of the center area of monolayer Ti₃C₂T_x MXene on a PTP device, there are only a few discontinuous Pt particles on the sample surface (Figure R7).

Notably, it is widely acknowledged that the sprayed Pt particles exhibit a soft character while Ti₃C₂T_x nanosheets have strong and brittle mechanical properties, so the propagation of Pt will not exert a substantial effect on the mechanical properties of MXene (*Acta Materialia* 2012, 60, 2258; *Nano Letters* 2011, 11, 3207). Furthermore, we have experimentally verified that Pt deposition didn't affect the mechanical properties of monolayer Ti₃C₂T_x nanosheets, evidenced from the data summarized in Table 1 below (Sample #4 and #5 have low Pt depositions showed similar results with the #1-#3 samples that have high Pt deposition).

Figure R7. TEM images of the center area of monolayer $Ti_3C_2T_x$ MXene on a PTP device with corresponding SAED pattern.

“As previously mentioned, the Pt deposited onto the surface of monolayer $Ti_3C_2T_x$ nanosheets during testing has a soft nature and won't impact the mechanical properties of the strong and hard $Ti_3C_2T_x$ MXene. The results are additionally validated via experimental methods. Initially, high-energy Pt deposition is employed to fix both ends of $Ti_3C_2T_x$, resulting in the sample becoming opaque under SEM (**Figure S10**) due to Pt deposition on the sample surface, and the measured mechanical properties are shown in **Table 1** (Sample #1-3). Subsequently, by setting the Pt deposition to low energy, the sample maintains the transparent nature, and the measured mechanical properties are presented in **Table 1** (Sample #4-5).”

Table 1. Mechanical properties of monolayer $Ti_3C_2T_x$ nanosheets. The displacement measurements by SEM itself should add ~7% uncertainty, the uncertainties in the measured values of Young's modulus and tensile strength are 2.7% and 12.5% respectively, which are within reasonable limits.

Sample #	Length (μm)	Width (μm)	Young's modulus E_{3D} (GPa)	Ultimate tensile strain (%)	Tensile strength (GPa)
1	2.5	5	488.2	2.8 ± 0.196	13.7 ± 0.96
2	2.5	5	469.2	3.1 ± 0.217	14.5 ± 1.02
3	2.5	5	475.6	3.2 ± 0.224	15.2 ± 1.07
4	2.5	5	494.6	3.6 ± 0.252	17.8 ± 1.25

5	2.5	5	489.8	3.2 ± 0.224	15.7 ± 1.10
Average			483.5 ± 13.2	3.2 ± 0.652	15.4 ± 1.92

10. After mounting the monolayer nanosheet, it was cut into desired shape and geometry using FIB. During this process, the entire specimen was likely exposed to the ion beam (from imaging, positioning of the milling pattern, etc.). Since the specimen is a monolayer with a thickness of 5 atomic layers, the influence of the Ga⁺ ion beam on the specimen would be quite significant (specimen damage, Ga implantation). Would this be a contributing factor to the lower observed tensile strength compared to the theoretical value, in addition to the edge defects claimed by the authors?

Response: We thank the reviewer for the constructive suggestion. The crystal nature of the tested Ti₃C₂T_x can be confirmed by a series of SAED patterns from the edge to the center area in **Figure S6a-c**. The fracture edges area of the tested Ti₃C₂T_x sample on the PTP device was further analyzed by TEM, as shown in **Figure S6a**, and the SAED pattern demonstrates that the crystal structure of the tested sample remained unchanged. The TEM in **Figure S6b** shows the edge area of the tested sample on PTP after being cut by FIB, and Ga⁺ causes localized inhomogeneous sputtering at the sample edges even though the minimum current (1pA) has been set, corresponding SAED confirms that the crystal structure remains unchanged after FIB. **Figure S6c** shows a TEM image of the center area of the tested Ti₃C₂T_x MXene on the PTP, which found no effects of Ga⁺ sputtering and only a small amount of discontinuous Pt residue, and the corresponding SAED indicates the high quality of sample. It is worth noting that if there are a large number of defects in the 2D materials' internal region, this will obviously modulate the fracture behavior, resulting in multiple crack stages. Furthermore, the fracture strength of the sample with edge defects remains in the same order of magnitude as the ideal strength of the defect-free sample, and the experimental measurements exceed half of the ideal value (i.e., deep ultra-strength). These effectively demonstrate that the effect of FIB on the fracture strength of monolayer Ti₃C₂T_x nanosheets is confined to the edge area only. In addition, this phenomenon has also been demonstrated in other studies using FIB to treat 2D thin film materials (Nat. Commun. 2020, 11, 284; Adv. Mater. 2017, 29, 1604201). Therefore, the effect of FIB on MXene flakes only causes edge defects, and the structure of the Ti₃C₂T_x flakes on the PTP device was unchanged during the transferring process and maintained the original properties.

“Given the occurrence of Pt propagation and localized sputtering of Ga⁺ during the experimental procedure, it is necessary to provide a comprehensive analysis and elucidation of the effects of Pt and Ga⁺ on the mechanical test results of the samples. As previously mentioned, the Pt deposited onto the surface

of monolayer $Ti_3C_2T_x$ nanosheets during testing has a soft nature and won't impact the mechanical properties of the strong and hard $Ti_3C_2T_x$ MXene. The results are additionally validated via experimental methods. Initially, high-energy Pt deposition is employed to fix both ends of $Ti_3C_2T_x$, resulting in the sample becoming opaque under SEM (**Figure S10**) due to Pt deposition on the sample surface, and the measured mechanical properties are shown in **Table 1** (Sample #1-3). Subsequently, by setting the Pt deposition to low energy, the sample maintains the transparent nature, and the measured mechanical properties are presented in **Table 1** (Sample #4-5). In addition, the SAED pattern of **Figure S6a** and **S6c** indicate that the properties of the tested $Ti_3C_2T_x$ MXene are unchanged, and the comparison of experimental results confirms that the Pt deposition didn't exert a significant impact on the test. During the FIB cutting process, we set a minimum current of 1 pA to minimize the edge defect concentration. As can be seen from **Figure S6a-c**, the effect of Ga^+ on the samples was limited to the cut edges, and the crystal nature of the $Ti_3C_2T_x$ MXene was unchanged. It is worth noting that if there are a large number of defects in the 2D materials' internal region, which will obviously modulate the fracture behavior and result in multiple crack stages⁵⁵. Furthermore, the fracture strength of the sample with edge defects remains in the same order of magnitude as the ideal strength of the defect-free sample, and the experimental measurements exceed half of the ideal value (i.e., deep ultra-strength⁵⁶). These effectively demonstrate that the effect of FIB on the fracture strength of monolayer $Ti_3C_2T_x$ nanosheets is confined to the edge area only. In addition, this phenomenon has also been demonstrated in other studies using FIB to treat 2D materials^{28,39,57}.

Figure S6. TEM and STEM images of monolayer $Ti_3C_2T_x$ MXene on a PTP device. (a, b, c) The crystal structure of tested sample on the PTP device: Low-magnification TEM image of the fracture edges, the FIB-cut edges, the center area of $Ti_3C_2T_x$ MXene, and corresponding SAED pattern, respectively. (d) Low-magnification STEM image of the fracture edges area of $Ti_3C_2T_x$ MXene. Zoom-in view of the red rectangle area shows the fractured cross-sectional surface of the tested sample.

11. The authors claimed that by using a lower FIB milling current, defect concentration at the edge of the sample can be reduced, resulting in higher fracture strength of $Ti_3C_2T_x$. However, there is no experimental evidence presented in the manuscript to support this claim.

Response: Thank you very much for your comment. We cut the $Ti_3C_2T_x$ nanosheets by setting FIB currents of 1 pA and 20 pA respectively. As can be seen from **Figure S9**, the high current of 20 pA not only causes high-concentration large-size defects on the edges of the nanosheet, but also results in large-scale Ga^+ sputtering, and even causes defects in areas far from the edges. The combination of common sense and our simulation results can effectively

support our claim.

Figure S9. The effect on $Ti_3C_2T_x$ MXene nanosheets edge defect concentration by setting different FIB currents and voltages to cut the sample: (a) TEM images of the edges area of tested sample cut by FIB setting 2 kV, 1 pA. (b) TEM images of the edges area of the sample cut by FIB setting 2 kV, 20 pA.

REVIEWERS' COMMENTS

Reviewer #1 (Remarks to the Author):

The authors have fully responded for my comments. I believe that the manuscript has been substantially improved and can be published in Nature Communications journal.

Reviewer #2 (Remarks to the Author):

The authors have adequately addressed my comments.

Reviewer #3 (Remarks to the Author):

Considering all the changes made to the manuscript after the first round of the review process, the quality of the manuscript was significantly improved.

I found especially useful extra information about the thickness of the MXene flakes, Pt deposition, and other new details regarding the application of FIB for the sample preparation.

Although, I am still sceptical about the authors' conclusions about the monolayer character of all tested flakes (I think high-resolution STEM/TEM imaging of the flakes on a PTP device before tensile tests and comparison of these images with simulated TEM/STEM images for monolayer MXene would more convincing), I believe the manuscript deserves publishing in Nature Communication.

Reviewer #4 (Remarks to the Author):

The quality of the manuscript is improved after the revision. The authors have addressed my previous questions. Below are additional comments and questions.

1. It is great that the parameters for STEM are now included. However, critical parameters including probe current (in pA) and electron dosage (in $e^-/\text{\AA}^2$) are still missing.

2. Some of the peaks in the survey scan X-ray photoelectron spectrum (Figure S3) are still not labelled.

Response to Reviewers

Reviewer #1

The authors have fully responded for my comments. I believe that the manuscript has been substantially improved and can be published in Nature Communications journal.

Response: The authors would like to thank the reviewer for the careful review and acceptance of our article.

Reviewer #2

The authors have adequately addressed my comments.

Response: We thank the reviewer for the positive recommendation for our article.

Reviewer #3

Considering all the changes made to the manuscript after the first round of the review process, the quality of the manuscript was significantly improved. I found especially useful extra information about the thickness of the MXene flakes, Pt deposition, and other new details regarding the application of FIB for the sample preparation. Although, I am still sceptical about the authors' conclusions about the monolayer character of all tested flakes (I think high-resolution STEM/TEM imaging of the flakes on a PTP device before tensile tests and comparison of these images with simulated TEM/STEM images for monolayer MXene would more convincing), I believe the manuscript deserves publishing in Nature Communication.

Response: We sincerely thank the reviewer for the careful review and great recommendation for publication.

Reviewer #4

The quality of the manuscript is improved after the revision. The authors have addressed my previous questions. Below are additional comments and questions.

Response: We appreciate the careful review and constructive suggestions on our manuscript. The scientific nature of this paper was improved by absorbing the

suggestions. Below are addressed the additional comments point by point:

1. It is great that the parameters for STEM are now included. However, critical parameters including probe current (in pA) and electron dosage (in $e^-/\text{\AA}^2$) are still missing.

Response: We thank the reviewer for this comment. Ga^+ FIB was used for transferring and milling the flake for a bone-like shape. In order to minimize irradiation damage during FIB cutting, a small probe current of 1 pA and extra low accelerating voltage of 2 keV were set. This is within a reasonable range, because the FIB current for cutting monolayer 2D materials is generally set at 1 pA and 2 keV. We have also added these details to the manuscript:

“In order to minimize irradiation damage during FIB cutting, an extra low accelerating voltage of 2 keV and small probe current of 1 pA were set.”

2. Some of the peaks in the survey scan X-ray photoelectron spectrum (Figure S3) are still not labelled.

Response: We thank the reviewer very much for the comments. We have labeled all peaks of X-ray photoelectron spectrum (XPS) in Supplementary Fig. 3, as shown below:

Name	Peak BE	Height CPS	FWHM eV	Area (P) CPS.eV
Ti 2p	455.85	243933	3.54	1335697.77
O 1s	530.05	96897.72	3.28	368686.88
F 1s	685.02	88590.1	3.03	360617.09
C 1s	282.37	55519.51	2.95	264481.58
Cl 2p	199.44	15644.18	3.59	70262.91

Supplementary Fig. 3 Full XPS spectrum of the prepared monolayer $Ti_3C_2T_x$ MXene. F KLL, O KLL and C KLL are the Auger electron peaks of F, O and C. Ti LMM and Ti LMMI are the two Auger electron peaks of the Ti.